# Ultrafast coherent control of a hole spin qubit in a germanium quantum dot

Ke Wang [1,2,8], Gang Xu[1,2,8], Fei Gao[3], He Liu[1,2], Rong-Long Ma[1,2], Xin Zhang[1,2], Zhanning Wang[4], Gang Cao[1,2], Ting Wang [3], Jian-Jun Zhang [3✉], Dimitrie Culcer [4], Xuedong Hu[5], Hong-Wen Jiang[6], Hai-Ou Li [1,2✉], Guang-Can Guo[1,2] & Guo-Ping Guo [1,2,7✉]

Operation speed and coherence time are two core measures for the viability of a qubit. Strong spin-orbit interaction (SOI) and relatively weak hyperfine interaction make holes in germanium (Ge) intriguing candidates for spin qubits with rapid, all-electrical coherent control. Here we report ultrafast single-spin manipulation in a hole-based double quantum dot in a germanium hut wire (GHW). Mediated by the strong SOI, a Rabi frequency exceeding 540 MHz is observed at a magnetic field of 100 mT, setting a record for ultrafast spin qubit control in semiconductor systems. We demonstrate that the strong SOI of heavy holes (HHs) in our GHW, characterized by a very short spin-orbit length of 1.5 nm, enables the rapid gate operations we accomplish. Our results demonstrate the potential of ultrafast coherent control of hole spin qubits to meet the requirement of DiVincenzo's criteria for a scalable quantum information processor.

[1] CAS Key Laboratory of Quantum Information, University of Science and Technology of China, 230026 Hefei, Anhui, China. [2] CAS Center for Excellence and Synergetic Innovation Center in Quantum Information and Quantum Physics, University of Science and Technology of China, 230026 Hefei, Anhui, China. [3] Institute of Physics and CAS Center for Excellence in Topological Quantum Computation, Chinese Academy of Sciences, 100190 Beijing, China. [4] School of Physics, University of New South Wales, Sydney 2052, Australia. [5] Department of Physics, University at Buffalo, SUNY, Buffalo, NY 14260, USA. [6] Department of Physics and Astronomy, University of California, Los Angeles, CA 90095, USA. [7] Origin Quantum Computing Company Limited, 230026 Hefei, Anhui, China. [8] These authors contributed equally: Ke Wang, Gang Xu. ✉email: jjzhang@iphy.ac.cn; haiouli@ustc.edu.cn; gpguo@ustc.edu.cn

Perfecting the quality of qubits hinges on high fidelity and fast single- and two-qubit gates. Electron spin qubits in Silicon (Si) quantum dots (QD) are considered promising building blocks for scalable quantum information processing[1–5], with long coherence times and high gate fidelities already demonstrated[6–14]. The conventional approach of using magnetic fields to operate single-qubit gates results in relatively low Rabi frequencies[12], spurring the development of electrically driven spin resonance based on the spin–orbit interaction[15] as an alternative. This all-electrical approach promises faster Rabi rotations and reduced power consumption, as well as paving the way towards scalability since electric fields are much easier to apply and localize than magnetic fields. In a Si QD, with relatively weak intrinsic SOI, a synthetic SOI has been introduced to provide fast and high-fidelity gates: a Rabi frequency above 10 MHz and gate fidelity of 99.9% have been reported in an isotopically enriched dot[6]. However, the magnetic field gradient enabling the synthetic SOI also exposes the system to charge-noise-induced spin dephasing[6,16], posing a formidable technical challenge. As such, the search for a high-quality spin qubit with fast manipulation and slow decoherence remains open[17].

Hole spins provide an intriguing alternative for encoding qubits as compared to conduction electrons[18–21], in particular in group IV materials such as Si and Ge[22–34]. Thanks to their underlying atomic P orbitals, which carry a finite angular momentum and have odd parity, holes experience an inherently strong SOI and weak hyperfine interaction[35,36]. The strong spin-orbital hybridization of hole states opens the door to fast all-electrical spin control. To date, several works on hole spin qubits have been reported in a multitude of systems, such as Si metal-oxide-semiconductor (MOS)[37], undoped strained Ge quantum well[38,39] and GHW[40] structures, with Rabi frequencies in the range of 70–140 MHz. A recent experiment in Ge/Si core/shell nanowire has reached a very fast Rabi frequency of 435 MHz, with a short spin-orbit length of 3 nm[41].

Here we advance the ultrafast control of hole spin qubits by performing faster spin rotations than any reported to date. By applying microwave bursts to one gate of a GHW hole double quantum dot (DQD)[31] and utilizing Pauli spin blockade (PSB) for spin-to-charge conversion and measurement, we observe multiple electric dipole spin resonance (EDSR) signals in the DQD. At one of these resonances, we achieve a Rabi frequency exceeding 540 MHz, with a dephasing time of 84 ns and a quality factor of ~ 45. This ultrafast driving is enabled by a very strong SOI, with an equivalent hole spin–orbit length of 1.5 nm. The driving speed is a strong function of the EDSR peaks we study, hence even higher quality factors are likely as qubit encoding is optimized.

## Results

**Measurement techniques and EDSR spectrum.** A scanning electron microscope (SEM) image of the DQD device is shown in Fig. 1a (Supplementary Fig. 1a shows a schematic of the device). The device consists of an insulating layer and five electrodes above a GHW grown using the Stranski-Krastanow (S-K) method[42,43]. The charge stability diagram of the DQD is mapped out and given in Supplementary Fig. 1b, with a zoom-in to one particular triple point given in Fig. 1b. Charge occupations of the DQD are about 5 holes in the left dot and 10 holes in the right dot. When measuring the current through the DQD at a d.c. bias of 3 mV (Fig. 1b and Supplementary Fig. 1c) and −3 mV (Supplementary Fig. 1d), a clear signature of PSB is observed: The zero-detuning current drops to 1 pA in the forward biased ($V_{sd} = 3$ mV), blocked configuration (dash base line of the triangle), compared to 30 pA ($V_{sd} = 3$ mV) in the reversed biased, non-spin-blocked regime. While PSB[44] is

usually detected in the (0,2) or (2,0) to (1,1) charge configurations, it has been observed in other charge configurations as well[45]. With this in mind, we conjecture that the transition we observe occurs near the (n + 1, m + 1) to (n, m + 2) charge transition, which can be equivalently described in terms of two-hole states near the (1,1) to (0,2) transition.

In the PSB regime, with a magnetic field **B** perpendicular to the substrate, we generate EDSR by applying a microwave pulse to gate R. The a.c. electric field displaces the hole wave function around its equilibrium position periodically, leading to spin rotation mediated by the strong SOI (Fig. 1d). When the microwave frequency matches the resonant frequency of the spin states and causes spin flips, PSB is lifted and an increase in the transport current is observed (a pure orbital transition without spin flip cannot lift the PSB and cannot affect the current). By mapping out the current as a function of **B** and microwave frequency $f$, we find multiple spin resonances, as shown in Fig. 1c and Supplementary Fig. 2.

The major observed resonances in Fig. 1c are well described by a two-hole model built upon a single singlet in the (0,2) charge configuration ($S_{02}$) and two-spin states $|↓↑⟩$, $|↑↓⟩$), $|↑↑⟩$ and $|↓↓⟩$ in the (1,1) charge configuration (Supplementary Note 2), as evidenced by Fig. 1e, f. The large number of resonances and different slopes in Fig. 1c are clear hints of different $g$-factors for the two dots. Indeed, to generate the theoretical spectrum in Fig. 1f, we use $g$-factors of 7 and 3.95 for the two dots. With such different $g$-factors[30,31,46], the two-spin states in the (1,1) regime should be spin product states for any magnetic field above 0.1 T. In the following, we focus on two of these resonances, denoted as mode A and mode B in Fig. 1c. Within our model, the corresponding transitions involve single-spin-flip in the left (A, between $|↓↓⟩$ and $|↑↓⟩$) and the right (B, between $|↓↓⟩$ and $|↓↑⟩$) dot.

**Rabi oscillations.** To demonstrate coherent control of a hole spin qubit, we apply a three-step pulse sequence on gate R (Fig. 2a) to generate Rabi oscillations for mode A at $B=100$ mT. The probability of a parallel spin state (spin blocked) or anti-parallel state (unblocked) is measured by the averaged current through the DQD as a function of the microwave burst duration $\tau_{burst}$ and microwave frequency $f$ (Fig. 2b). We can resolve up to seven oscillations within 180 ns, and the standard chevron pattern helps us to pinpoint the qubit Larmor frequency at 7.92 GHz. To investigate how fast the qubit can be driven coherently, we vary the microwave power $P$ of the driving field from 0 dBm to 9 dBm and measure the Rabi frequency of mode A (Fig. 2d). Rabi oscillations at $f = 7.92$ GHz with a fit to $A \cdot \cos(2\pi f_{Rabi}\tau_{burst} + \varphi) \cdot \exp(-(\tau_{burst}/T_2^R)^2) + I_0$ (An offset of 0.5 pA is set between two oscillations for clarity) are shown in Fig. 2c at three different microwave power $P = -5, 0, 6$ dBm. At the strongest driving with $P = 9$ dBm (Fig. 4b), we achieve a Rabi frequency of $f_{Rabi} = 542 \pm 2$ MHz for mode A and $291 \pm 1$ MHz for mode B.

**Free evolution and decoherence.** Decoherence determines the quality of the hole spin qubit. To evaluate the dephasing time $T_2^*$ for mode A, we perform a Ramsey fringe experiment[6,7,37–40], with the pulse sequence shown in the top panel of Fig. 3f. A pattern of Ramsey fringes is shown in Fig. 3a when we vary the waiting time $\tau$ and microwave frequency detuning $\Delta f = f - f_0$ ($f_0 = 7.92$ GHz is the Larmor frequency). The fringes remain visible up to $\tau \sim 60$ ns, giving a qualitative indication of the dephasing time. We perform a Fast Fourier Transformation (FFT) of the Ramsey pattern in Fig. 3b, where the Ramsey oscillation frequency ($f_{Ramsey}$) equals $\Delta f$. Alternatively, two-axis control can be achieved by varying the relative phase $\Delta\varphi$ of the microwave modulation between the two pulses[6,15,37]. The results of relative phase ($\Delta\varphi$) in cycles identify the control of the

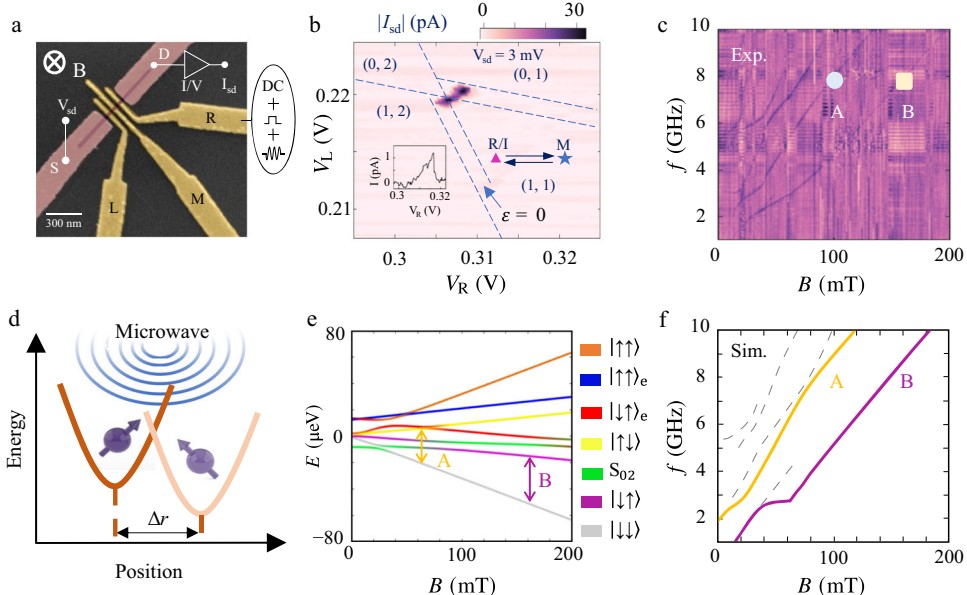

**Fig. 1 Experimental setup and EDSR spectrum. a** Scanning electron microscope image of the DQD device. Magentas metals are ohmic contacts while gates used to tune different potentials of DQDs are highlighted with yellow (L, M, and R). Gate L (R) produces the confinement potential for the left (right) dot. The tunnel coupling between the two dots is controlled by gate M. Microwave pulses are applied via gate R. **b** A conductance triangle of the DQD at $V_{sd} = 3\,mV$. Due to the weak signal of the small current, we mark the transitions of the triangle by blue dashed lines while the arrow indicates the position of the detuning $\varepsilon = 0$ (the energy difference between the left dot and the right dot) at the base line of the triangle. A suppressed current of ~1 pA is observed, which can be lifted by spin resonance. Points R, I, and M mark the readout, initialization and manipulation positions respectively. Inset: a line trace at $V_L = 0.215\,V$. **c** EDSR spectrum, measured by applying a continuous microwave with a power of −15 dBm at the point R/I. The circle and square symbols show the working points for subsequent experiments, corresponding to a Larmor frequency ~8 GHz. **d** Schematic of spin-orbit-coupling-mediated spin flip: the microwave electric field generated by the gate creates oscillatory displacements of the hole wave function ($\Delta r$) and its energy. As described in our model, $\Delta r$ makes contribution to the energy shift mediated by the spin-orbit coupling $\Delta_{so}$. Such orbital dynamics lead to spin flip with the help of SOI. **e** Other related resonances are colored in grey in **f** (see details in Supplementary Note 2, EDSR spectrum). **f** Calculated EDSR spectrum from our effective two-hole model with $g$-factors of left and right dot of $g_L = 7$ and $g_R = 3.95$. The two highlighted resonances (purple, yellow) correspond to spin-flips between $|\downarrow\downarrow\rangle$ and $|\uparrow\downarrow\rangle/|\downarrow\uparrow\rangle$, indicated by the arrows in the energy level diagram **e**.

rotation axis in addition to $\Delta f$ (Fig. 3c). From the Ramsey experiment of mode A, dephasing times of $T_2^* = 84 \pm 9$ ns and $T_2^* = 42 \pm 4$ ns are extracted at $P = -10$ dBm (Fig. 3d) and $P = 9$ dBm (Fig. 3e), respectively. The former is a better representation of hole spin dephasing, while the latter reflects coherence degradation from the onset of microwave-induced heating. The measured coherence times can be extended by performing a Hahn echo pulse sequence. For instance, the coherence time is extracted to be $T_2^* = 66 \pm 6$ ns in mode B at $P = 0$ dBm (Fig. 3f), while an enhanced coherence time of 523 ns is obtained using Hahn echo (Fig. 3g).

**Spin–orbit coupling strength**. Unlike $z$-rotation speed, which is simply determined by the control microwave, i.e., frequency detuning, the $x$-rotation speed for the hole spins is determined by SOI strength together with the strength of the driving electric field[22]. With a three dimensional confinement (with $z$-direction confinement much stronger than the other two dimensions, $L_y = 40\,nm >> L_x >> L_z$) and an out-of-plane magnetic field, the lowest states in the subspace spanned by the spin-3/2 hole states can be calculated, yielding states that are 95% HH (Supplementary Note 9). With the relatively small number of holes per dot (5–10), we attribute the manipulated spin states to be HH throughout the manuscript. The EDSR signals we observe are thus mediated by the strong SOI of 2D heavy-holes.

The leading SOI for a 2D HH gas is the Rashba term[47]

$$H_{so} = i\alpha_2(k_+^3 \sigma_- - k_-^3 \sigma_+) \tag{1}$$

where $\sigma_\pm = (\sigma_x \pm i\sigma_y)/2$, $k_\pm = k_x \pm ik_y$ and the Rashba constant $\alpha_2$ arises from the spherical component of the Luttinger

Hamiltonian[48]. We have determined that the cubic-symmetry term $\propto \alpha_3$ is negligible in a nanowire, being an order of magnitude smaller (Supplementary Note 9). We have performed a single-hole model calculation with in-plane confinement of an asymmetric harmonic potential $V(x, y) = \frac{1}{2}m\omega_x^2 x^2 + \frac{1}{2}m\omega_y^2 y^2$. By projecting $H_{so}$ onto the eigenstates of the 2D harmonic oscillator we obtain the transition matrix element for Rabi oscillations:

$$hf_{Rabi} = g\mu_B B \cdot \frac{a_x}{l_{so}} \cdot \frac{eE_{ac}a_x}{\hbar\omega_y} \tag{2}$$

where $a_x = \sqrt{\hbar/(m^*\omega_x)}$ is the transverse QD size, $g$ the Lande $g$-factor for a single spin, $\mu_B$ the Bohr magneton, $B$ the static magnetic field, $m^*$ the Ge HH effective mass, $E_{ac}$ the effective electric field at the QD generated by the microwave pulses applied to gate R, and $l_{so} \propto 1/\alpha_2$ the spin-orbit length defined in the Supplementary Note 10.

We can obtain the spin–orbit coupling strength according to Eq. (2) from observations of Rabi oscillations of both mode B (Fig. 4a and Supplementary Fig. 7.1) and mode A (Fig. 4a and Supplementary Fig. 6) in the range of $P \leq 9$ dBm. In mode A, nine oscillation periods are observed within 16 ns at $P = 9$ dBm (Fig. 4b), with a Rabi frequency of $f_{Rabi} = 542 \pm 2$ MHz at $f = 7.92$ GHz. When the microwave power is further increased (Supplementary Fig. 6b, c), photon-assisted tunneling (PAT) limits the detection of coherent control at higher Rabi frequencies[6,7,15,19,20]. Replacing $E_{ac}$ (Supplementary Fig. 8) with $P$ in Eq. (2), spin-orbit lengths of 1.5 nm and 1.4 nm are obtained by fitting the linear dependence of $f_{Rabi}$ on $E_{ac}$ in mode A and B, respectively, corresponding to a strong spin-orbit coupling

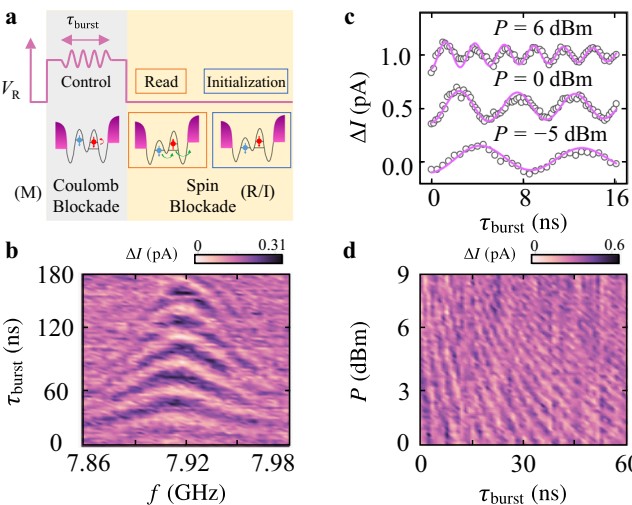

**Fig. 2 Ultrafast coherent spin control of mode A. a** Schematic representation of the spin manipulation cycle and corresponding gate-voltage ($V_R$) modulation pattern. The spin state is first initialized at point I in the PSB regime (Fig. 1b). A pulse then detunes the state to point M in the Coulomb blockade regime for spin manipulation. During this step, a microwave burst with a duration of $\tau_{burst}$ is applied to generate spin rotation via EDSR. Afterwards, the system is shifted from Coulomb blockade regime back to spin blockade regime at point R for readout. **b** At an external magnetic field of $B = 100$ mT, spin oscillation is observed by sweeping the microwave frequency $f$ and microwave duration time $\tau_{burst}$ (the amplitude from the pulse signal generator is set at $P = -15$ dBm) applied to the gate R. Each data point is averaged over 20 repetitions. **c** Rabi oscillations at $f = 7.92$ GHz with a fit to $A \cdot \cos(2\pi f_{Rabi}\tau_{burst} + \varphi) \cdot \exp(-(\tau_{burst}/T_2^R)^2) + I_0$ (An offset of 0.5 pA is set between two oscillations for clarity). Rabi frequencies are $112 \pm 2$, $202 \pm 2$ and $393 \pm 2$ MHz from bottom to top. We correct the data by removing the background current $I_0$. Similar results for mode B are shown in Supplementary Fig. 7.1. To mitigate the effects of charge noise, we average the current over 100 repeated cycles for each data point (Supplementary Fig. 5). **d** Rabi oscillations under different microwave power $P$.

strength of $\alpha_2 \sim 680$ meV·nm³. Due to the smaller a.c. the field in the right dot compared to the left dot (Supplementary Note 6) and the absence of a low-energy intermediate state (Supplementary Note 2), mode B shows a slower speed of Rabi rotation compared to mode A at the same microwave power even though the spin–orbit coupling strengths are similar. Notice that while spin–orbit length is a concept more appropriate for free carriers, it is nonetheless useful for comparing different systems. For example, for strongly spin-orbit coupled conduction electrons confined in InAs quantum dots of similar size to our dots, typical spin-orbit length ranges from 100 to 200 nm[49]. In comparison, our hole system has, inherently, a much stronger spin-orbit coupling (thus a much smaller $l_{so}$), which is the determining factor for the ultrafast operation of our qubit.

## Discussion

Ultrafast control of hole spins has also been achieved in a Ge/Si core/shell nanowire[41], though that hole system is quite different from ours. The key difference stems from the respective geometries. Our GHW has a confinement potential that is much stronger in one direction, akin to a quantum well, for which theory predicts a spin-3/2 (i.e. 'heavy hole') ground state. Ge/Si core/shell nanowires have cylindrical symmetry, where theory predicts a spin-1/2 ('light-hole') ground state[5]. Due to possible mass reversal, the magnitude of the effective mass is not a reliable indicator of spin character. On the other hand, the two systems

do share the feature of a strong spin–orbit coupling, which enables fast control.

The large Rabi frequencies in our system are achieved with a small driving electrical field $\sim 10^4$ V/m, compared to $\sim 10^6$ V/m static electrical field used for QD confinement (Supplementary Fig. 8). We thus attribute the large Rabi frequencies of both mode A and B to a large value of $\alpha_2$. The particularly large Rabi frequency in mode A may have been enhanced by the low-energy excited state included in our model (Supplementary Note 2 and 10), a fact that has previously been exploited to enhance spin-electric-field coupling[50,51]. To obtain a deeper understanding of such strong spin–orbit coupling, anisotropy spectroscopy would be a viable method to reveal the underlying mechanisms[46,52]. Even though the Rabi frequency of 540 MHz is limited by unwanted PAT and heating effects at high microwave power, we believe it is still not the upper bound for the Rabi frequency of a GHW hole spin qubit. For instance, faster operation is possible if another branch of EDSR with a larger Larmor frequency can be identified in the DQD. A particular example is the transition between $|\downarrow\downarrow\rangle$ and $|\uparrow\uparrow\rangle$, which can be observed in our system (brown in Supplementary Fig. 2a). This spin-flip transition corresponds to a larger energy splitting compared to mode A and mode B at the same magnetic field and could result in a larger Rabi frequency as EDSR frequency is proportional to Zeeman splitting. Moreover, faster Rabi oscillation can be achieved by changing the manipulation position. Our test results show that Rabi frequency can be increased from 63 MHz to 111 MHz by switching the manipulation position from M2 to M1 (Supplementary Fig. 4). We are thus optimistic that even faster Rabi operation is achievable after optimization.

A high-quality qubit requires both fast manipulation and slow decoherence. We have thus investigated dephasing for both modes A and B (Supplementary Fig. 6d and Fig. 3f). The dephasing rate appears approximately uniform across the two modes, so that the qubit quality factor $Q = 2f_{Rabi}T_2^{Rabi}$ is roughly determined by the Rabi frequency of the different modes. We thus obtain a lower bound estimate of the quality factor $\sim 45$ using $f_{Rabi} = 542$ MHz and $T_2^* = 42$ ns at $P = 9$ dBm. This value predicts a fidelity of the $\pi$ gate to be $e^{-1/Q} = 97.8\%$. A benchmark for Rabi frequency of QD spin qubits is set in Supplementary Fig. 10. Our hole spin qubit has one of the fastest Rabi frequency, with a quality factor around 45 that we believe can be further improved.

In conclusion, we have achieved ultrafast spin manipulation in a Ge HW. We report a Rabi frequency of up to 540 MHz at a small magnetic field of 100 mT, and obtain a dephasing time of 84 ns from a Ramsey fringe experiment. A hole spin qubit with a quality-factor of 45 is thus realized in our experiment. As dephasing appears to change little across different modes, higher-quality qubits could be achievable for state combinations with stronger spin–orbit coupling. We report a small spin-orbit length in a smaller Ge double quantum dot compared to existing work on GHW in the literature[40] with narrower electrodes. We attribute the ultrafast control of a hole spin qubit that we have observed to an overall strong spin-orbit coupling, possibly assisted by a nearby excited state, even though the relative smaller longitude dot size (along $y$) may have reduced the Rabi frequency. In other words, our results demonstrate that hole spins in GHW QDs are intriguing candidates for semiconductor quantum computing, providing the ability of all-electrical ultrafast control without the need for a micromagnet[13] or a co-planar stripline[14].

## Methods

**Device fabrication.** Our hut wire was grown on Si (001) by means of a catalyst-free method based on molecular beam epitaxy. A Ge layer (1.5 nm) was deposited by S-K growth mode on a Si buffer layer (100 nm). A 3.5-nm-thick Si cap was then grown on top of the Ge layer to protect the nanowire with a width of 20 nm

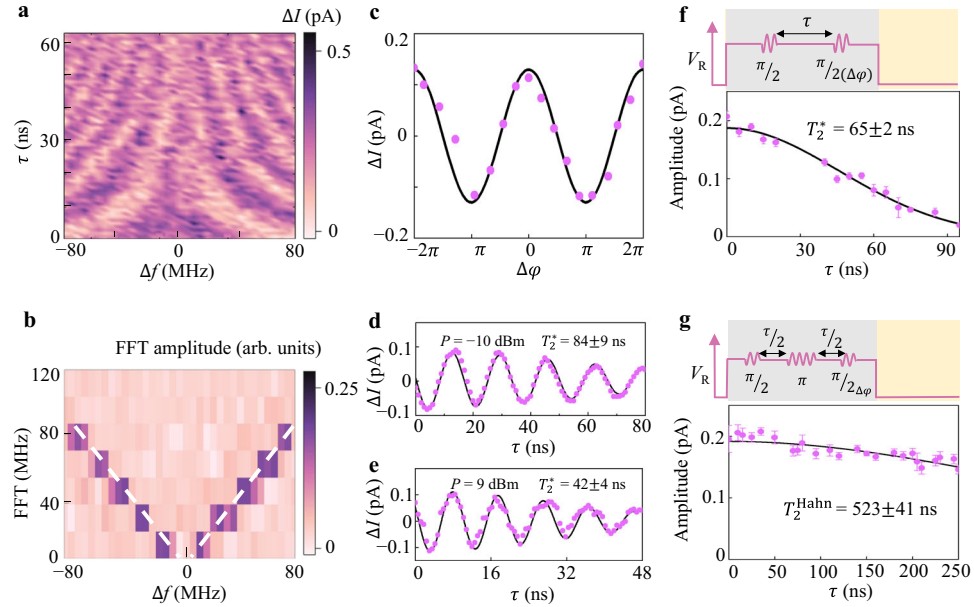

**Fig. 3 Free evolution and dephasing of the qubit. a** Ramsey fringes measured via transport current for mode A as functions of the driving microwave frequency detuning $\Delta f$ and free evolution time $\tau$ between two $\pi/2$ pulses at microwave power $P = -10$ dBm, Larmor frequency $f = 7.92$ GHz and magnetic field $B = 100$ mT, where the second pulse has the form of $E_{ac} \cos(2\pi f + \Delta\varphi)$. **b** Frequency FFT corresponding to the Ramsey fringes in **a**. Two white dash lines mark the dependence of $z$-axis rotation on the frequency detuning $\Delta f$. When $\Delta f = 0$, $f_{Ramsey} = 0$ as well, corresponding to a pure dephasing process. **c** Oscillations of the phase control of the second pulse at $\tau = 0$ ns and $\Delta f = 0$ MHz shows a perfect cycle of $2\pi$. When performing Ramsey experiment at the condition of $\Delta\varphi = 0$, $\frac{\pi}{2}$, $\pi$ and $\frac{3\pi}{2}$, the second pulse induces a rotation around the $x$, $y$, $-x$ and $-y$ axis, respectively. **d, e** Dephasing times for mode A at $T_2^* = 84 \pm 9$ ns and $T_2^* = 42 \pm 4$ ns are obtained from the decay of $\Delta I$ extracted from Ramsey experiment with a fixed $\Delta f$ by fitting $I = A \cdot \cos(2\pi f\tau + \varphi_0) \exp(-(\tau/T_2^*)^2) + I_0$ at $P = -10$ dBm and $P = 9$ dBm, respectively. **f** Pulse sequence used to control the free evolution of the qubit (top). Dephasing time for mode B: $T_2^* = 65 \pm 2$ ns at $P = 0$ dBm, $f = 7.88$ GHz, and $B = 156$ mT. **g** Hahn echo sequence (top). After the Hahn echo, dephasing time of the mode B qubit is extended to 523 ns. The duration of the pulses are $t_{\pi/2} = 2.5$ ns and $t_\pi = 5$ ns. Here we use $I = A \cdot \exp(-(\frac{\tau}{T_2^{Hahn}})^{1+\alpha})$ to fit our data, where $\alpha = 0.9$ is determined by the noise spectrum. (see Supplementary Fig. 5). All the s.d. error bars come from the fitting.

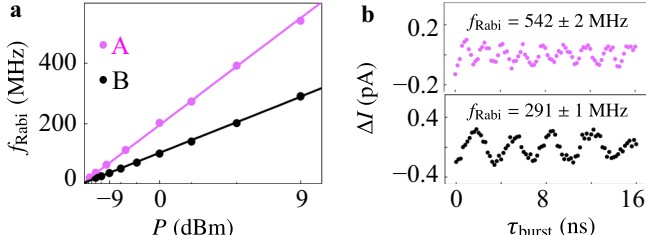

**Fig. 4 Rabi frequency and spin-orbit length. a** Linear dependence of Rabi frequency on the amplitude of applied driving field (i.e. $E_{ac}$). Rescale of power is converted to amplitude (i.e. $P^{1/2}$) by $P$ (mW) $= 10^{(P(dBm)-36)/10}$, $P^{1/2} = (2 \times P \text{ (mW)} \times 50\,\Omega)^{0.5}$. For $P > 9$ dBm, the data deviate from the linear dependence as a result of PAT effect (Supplementary Fig. 6c). Parameters for mode A are $a_x = 5$ nm, $\hbar\omega_y = 3$ meV (energy splitting between the first excited state and ground state in Fig. 1b), $g = 7$, $B = 100$ mT; Parameters for mode B are $a_x = 5$ nm, $\hbar\omega_y = 3$ meV, $g = 3.95$, $B = 156$ mT. Spin-orbit length of mode A and B are obtained to be 1.5 nm and 1.4 nm respectively, by fitting the Rabi frequency's dependence on the square root of power using Eq. (2). **b** Rabi oscillations of mode A (B) at microwave power $P = 9$ dBm after subtracting the linear background are shown in the top (bottom) panel.

(Details in Supplementary Fig. 1a). The dot size along $x$ direction of $a_x = 5$ nm is obtained based on the ground state calculation. The length of our wire can be longer than 1 μm, although a 500 nm wire is sufficient for our DQD device. On both ends of the wire, two 30-nm-thick palladium electrodes were defined as the ohmic contacts. A layer of aluminum oxide (25 nm) was then deposited on top of the nanowire and ohmic contacts as an insulator. Three 30-nm-wide Ti/Pd electrodes are deposited to generate the DQD potential in the wire.

**Experimental setup**. The experiments were performed in an Oxford Triton dilution refrigerator at a base temperature of 10 mK. The sinusoidal waves from the output of the arbitrary waveform generator Keysight M8190A are inputted to I/Q ports of the vector source Keysight E8267D in order to generate the modulated sequences for spin control. Combined with pulses from M8190A, these sequences are then transmitted by a semi-rigid coaxial line connecting to gate R, where the total attenuation is 36 dB. As shown in Fig. 2a, we apply two-stage pulses to gate R for spin initialization, control and readout. The length of one cycle is fixed at 640 ns and 320 ns of it is for spin control. The average transport current is measured by a digital multimeter after a low-noise current preamplifier SR570.

## Data availability

All the data that support the findings of this study are available from the corresponding author upon reasonable request.

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

## Acknowledgements

We acknowledge P. Huang for helpful discussions of EDSR spectrum. This work was supported by the National Key Research and Development Program of China (Grant No.2016YFA0301700), the National Natural Science Foundation of China (Grants No. 12074368, 92165207, 12034018, 61922074, and 11625419), the Anhui initiative in Quantum Information Technologies (Grants No. AHY080000), the Anhui Province Natural Science Foundation (Grants No. 2108085J03), the USTC Tang Scholarship and this work was partially carried out at the USTC Center for Micro and Nanoscale Research and Fabrication. H.-W.J. and X.H. acknowledge financial support by U.S. ARO through Grant No. W911NF1410346 and No. W911NF1710257, respectively. D.C. is supported by the Australian Research Council Future Fellowship FT190100062.

## Author contributions

K.W. performed the bulk of measurement and data analysis with the help of H.L. and G.X. fabricated the device. F.G., T.W., and J.-J. Z. supplied the Ge hut wires. K.W. wrote the manuscript with inputs from other authors. Z.N.W., D.C., and X.H., provided theoretical support and G.-C.G. and H.-W.J. advised on experiments. R.-L.M. contributed to the simulation and X.Z. and G.C. polished the manuscript. H.-O.L. and G.-P.G. supervised the project.

## Competing interests

The authors declare no competing interests.
