## [Peer Review File · Nature Communications]

REVIEWER COMMENTS

Reviewer #1 (Remarks to the Author):

The manuscript entitled “Ultrafast operation of a hole-spin qubit in a Germanium Quantum dot” reports the fast control of two hole spin modes at an unprecedented frequency, up to 540 MHz. This strong drive is explained by the intrinsically strong spin-orbit interaction of heavy holes in a germanium hut wire, with a spin-orbit length of 2.3 nm reported by the authors. In my opinion, this work represents an impressive technical achievement, and the experimental data support the claim of the paper. However, I believe several points would require a more thorough discussion in the manuscript, which is why I cannot recommend the publication in the current form. The two main questions I had concern the nature of the driven modes A and B, as well as the lack of perspective and explanation on the saturation of f_{Rabi} towards higher power.

In the main text, the authors mention that the two modes A and B “may correspond to spin flip transitions between the two QDs, or to spin transitions within the same QD”. In the supplementary figure S2, they present a rather complex diagram with 5 transitions of various slopes. Can this spectroscopy be used to provide hints on the nature of the two modes A and B? In particular, do the authors have an explanation for the important g-factor difference between these two modes (resp. 4.5 and 3).

Can the authors develop their explanation about the saturation of f_{Rabi} as P is increased above +9dBm? Would photon-assisted tunneling affect the spin manipulation or the readout? Is the T_2^* further reduced for P=10, 11 and 12dBm? Could these limitations be mitigated by changing the manipulation position M or lowering the interdot tunnel-coupling during the MW application?

I feel that the paper lacks perspective on a future improvement of this work. The authors mention that they “believe it is not the upper bound for the Rabi frequency of a GHW hole spin qubit”. Besides finding another mode with a stronger coupling to the MW drive, can the authors propose a strategy to further enhance the quality factor of their spin manipulation?

The paper by Froning et. al, discussed in the note at the end of the main text (ref [31]), should be added to the benchmark presented in Fig. 4 of the main text.

In the supplementary material, the labels of the benchmark (Fig. S9) are incorrect, they should be shifted by 1.

While the main text presents a good English level and clear explanations, the supplementary material contains many mistakes and missing words, with some sentences hardly understandable at all. For example, the caption of Fig. S3 is extremely unclear.

Reviewer #2 (Remarks to the Author):

GENERAL CONCLUSIONS:

The manuscript “Ultrafast Operation of a Hole Spin Qubit in a Germanium Quantum Dot” by Wang et al. presents a thorough study of hole spin qubits in germanium hut-wires. The data convincingly show fast qubit manipulation, and clear extraction of parameters such as coherence times and Rabi frequencies. The interpretation of the data is very likely to be true, but I find the current presentation below standard for publication:

1. The manuscript does not give appropriate credit to other works in the field, esp fig 4b (=fig S9) must be adapted, since it has errors and omissions
2. The statements about heavy / light hole behaviour are confusing
3. The interesting EDSR pattern deserves more attention
4. The nature of modes A and B is left vague
5. Ref [31] must get the credit it deserves.

The suggested revisions are below in my comments.

MAJOR COMMENTS

1. Fig 4b / S9 are in principle a very nice overview to compare this work to results from literature. However, there are 3 major concerns:
 - a. Is this a relevant way to plot it? If you want to plot the operation speed versus coherence, then the relevant coherence is the T_2 (instead of T_2^*). If you are interested the number of operations before decoherence, then that would be a more suitable benchmark. Even better would be the fidelity via randomized benchmarking, as in nearly all references.
 - b. Is this a fair way to plot it? Are all T_2^* values measured at the corresponding Rabi frequency? This does not seem to be the case for the data point of this work, while it does seem to hold for some of the literature values. That would be an unfair comparison.

c. Important values are missing (ref S17, and ref 31), and the Rabi frequency of S14 is placed below 10 MHz while they reach 100 MHz. Since there are some errors, I would suggest to double check all values by two separate authors.

2. The current version of the manuscript is unclear about the nature of the holes: whether they are heavy or light holes, or a mix. The different statements are spread through the manuscript and are confusing:

p1: "the strong SOI of heavy holes in our GHW"

p2: "...the lowest-energy levels are nearly pure heavy holes (HHs) with a light hole (LHs) admixture below 1%, although larger dot occupation could increase this admixture. Due to the hole spin-3/2, the multiplicity of available HH and LH spin states leads to a complex pattern of spin transitions,"

p3: "...the uncertainty in the hole occupation number and the possibility of mixing between LH and HH states, .."

p5: "Because of the relatively small number of holes per dot (~10) we assume EDSR is mediated by the strong SOI of 2D heavy-holes."

p5: " m^* the Ge HH effective mass,"

p5: "The spin-orbit length difference between modes A and B comes from different states involved in the respective transitions,"

a. I would advise to combine these in 1 paragraph, and/or add a discussion at the end of the manuscript about what can/cannot be concluded/speculated. HH-LH physics is a hot topic in our field, so a concise discussion will add value to the manuscript.

b. The estimate of the hole number should be substantiated with data and arguments. Is it similar for the two quantum dots?

3. The complex pattern of EDSR transitions in figure S2 is very interesting and should be explained clearer. This a great opportunity to gain a deeper understanding of HH and LH spins

a. At least a speculation of the origin of the EDSR lines and anti-crossings should be added to the manuscript.

b. The authors assign g-factors of 4.5 and 3 to mode A and B. I cannot retrieve where in the manuscript they write how they found these numbers. It looks like they used the f and B where the measurements were performed (~8 GHz). I am not sure whether this is justified, because that would imply that the g-factor strongly changes as the magnetic field is reduced to the anti-crossings (~2-3

GHz), e.g. mode A has $f \sim 3$ GHz at $B \sim 20$ mT, corresponding to $g \sim 10$. Following the same argument, the mode with $f \sim 6$ GHz at $B \sim 10$ mT would have $g \sim 43$. I would expect some discussion/explanation in the manuscript about the g -factors: Why these values? Why so different for two similar QDs? Why do they change strongly with B ? how are the values coupled to the picture of HHs, LHs and possible mixing?

In comments 2 and 3 above, I understand that many questions regarding g -factors and HH-LH cannot be answered based on the data. This is exactly why it is so important to at least clearly state the factual observations, what can be concluded and discuss possible explanations / speculations for what cannot be concluded. A clear distinction between what can and cannot be concluded can help the field to follow-up with new theory and experiments.

4. Page 5: "The spin-orbit length difference between modes A and B comes from different states involved in the respective transitions,". Do modes A and B correspond to the left and right quantum dot? If not, what could explain the modes (speculation is allowed)? Adding plot of T_2^* for mode A and mode B versus a useful parameter (e.g. f or f_{Rabi} or B or power) can help the reader. The origin of the differences between mode A and B must be discussed in the manuscript, similar to the 2 dots, perhaps in same paragraph.

5. Page 7: "While writing this letter, we became aware of a related preprint in which ultrafast control of hole spins is achieved in Ge/Si core/shell nanowire[31], which is quite different from our work". Reference 31 may have been noticed during writing, but is so close to this work that it cannot be mentioned only as a final note. I agree that in details the work is different, but a journal like Nature Communications deserves the big picture to be painted. The works are too similar to ignore in the main text. The similarities are (i) the Ge/Si material system (ii) the ultrafast operation (540 MHz versus 435 MHz Rabi) and (iii) short spin-orbit length (2.3 nm versus 3 nm). This manuscript still has the record, and must be published with due credit for ref [13], meaning not in an added note.

MINOR COMMENTS

6. I do not understand the difference between the top plot in figure 2d (Rabi frequency of 542 MHz at $P = 9$ dBm and $f = 7.92$ GHz) and figure 3e (Rabi frequency of ~ 100 MHz (my estimate) at $P = 9$ dBm and $f = 7.92$ GHz (?)). Figure 4a implies that a Rabi frequency of ~ 100 MHz is found at $P \sim -5$ dBm.

7. Figure 1:

- a. I find the dashed lines in fig 1b dangerously suggestive and would prefer the data without the dashed lines, or less pronounced. The reader should be able to judge by him/herself, which is not possible in the current version with thick overlaid lines.
- b. The caption of 1b refers to a conductance triangle, which is absent in the data
- c. The caption of 1b refers to “a suppressed current of $\sim 1\text{pA}$ at detuning ~ 0 , highlighted by the base of the triangle”. The detuning (axis) is not explained/defined, and the $\sim 1\text{ pA}$ is not visible, at least nothing stands out from the background coulomb blockade. A line trace may help.
- d. Fig 1c does not explain EDSR adequately
- e. The caption of 1d refers to “a continuous microwave with a power of -15 dBm .” Perhaps the authors can add that this is applied at point R/I in fig 1b?

8. Figure 2a:

- a. It may help to add the positions M and R/I above the left and right panels
- b. I find the 2 cartoons in R/I unclear. Sequential or 2 different situations? Should be made clearer

9. Figure 3f-g: the number of significant digits in $65 \pm 2\text{ ns}$ in 3f is different from the number in $523 \pm 41\text{ ns}$ in 3g. Is this justified?

Floris Zwanenburg

Reviewer #3 (Remarks to the Author):

The manuscript submitted by Wang et al. presents an experimental study about the fast driving of a spin qubit located in a germanium quantum dot.

The QD device is fabricated based on a germanium hut wire with top gates to localize quantum dots along the wire. Wang et al. uses Pauli spin blockade in a multi-hole regime to read-out spin-orbit states manipulation. The authors have studied two electrically driven spin resonance (ESDR) and have shown for one of the resonance an ultra-fast Rabi frequency of 542MHz . They also studied the free evolution and the dephasing of this ultra-fast resonance. Finally, Wang et al. attributes the ultra-fast Rabi frequency to an unusually short spin-orbit length of 2.3nm .

While the experimental results are appealing at a first glance, the theoretical explanation given by the authors about the ultra-fast Rabi oscillation due to an unusually short spin-orbit length seems difficult to apply to the experimental results (see following explanation)

Moreover a spin-orbit qubit in a Ge hut wire has already been demonstrated by Watzinger et al. in 2018 and published in Nature Communications.

Considering these two last aspects I cannot recommend the publication of the present manuscript in Nature Communications.

Explanation:

From Fig.S2, the EDSR spectrum is complex as it presents a lot of resonances, which indicates that the simple picture of a $(1,1)/(0,2)$ transition cannot be applied here (certainly low level splitting are present in one of the dots). Consequently it is hard to know which states are driven for the EDSR A and B.

Still from Fig.S2 the EDSR line A seems to anti-cross with the EDSR line just above, indicating that one of the states involved in the EDSR A is interacting with another level of the double dot.

Consequently EDSR A is not associated to the driving of a simple spin splitted orbital state in one of the dot.

The authors indicates a g -factor of 4.5 and 3 for the EDSR A and B, respectively. To my knowledge, these g -factor are far from the heavy-hole g -factor expected in Germanium.

From all the above remarks, I do not understand how the authors can claim (even qualitatively) that they are driving an HH spin transition. From the complexity of the EDSR spectrum, it appears clear that the level splitting in one of the QD (at least) is small. From the small g -factor, it is also clear that the states involved in the EDSR resonances have mixed heavy and light hole character. The Rabi frequency observed is then certainly coming from a complex interplay between spin-orbit interaction and multi-level structure with heavy and light admixture of the QD under consideration.

Consequently, I cannot agree with the authors on the description of the EDSR mechanism has just the effect of a Rashba-type SOI acting on spin $3/2$ heavy hole.

Response to Reviewers' Comments

We thank the reviewers for their thorough reading and insightful comments on how to improve the manuscript. In particular, we have made the physical picture more clear by explaining our experimental observation of the ultrafast control of a heavy-hole spin-orbit qubit with a two-hole model. The point-by-point responses to the Reviewers are detailed below. The comments from the referees are reproduced in *purple color and Calibri font* along with our responses in black text. Modified contents are indicated by *blue color and Times New Roman font* in the revised main manuscript and revised Supplementary Material.

Response to Reviewer 1

Comments: The manuscript entitled "Ultrafast operation of a hole-spin qubit in a Germanium Quantum dot" reports the fast control of two hole spin modes at an unprecedented frequency, up to 540 MHz. This strong drive is explained by the intrinsically strong spin-orbit interaction of heavy holes in a germanium hut wire, with a spin-orbit length of 2.3 nm reported by the authors. In my opinion, this work represents an impressive technical achievement, and the experimental data support the claim of the paper. However, I believe several points would require a more thorough discussion in the manuscript, which is why I cannot recommend the publication in the current form. The two main questions I had concern the nature of the driven modes A and B, as well as the lack of perspective and explanation on the saturation of f_{Rabi} towards higher power.

Response: We are grateful for the positive comments of the Reviewer and acknowledgement of our technical achievement, and we thank the Reviewer for his/her insightful and valuable comments to improve our work. We address each of these comments below.

- 1. In the main text, the authors mention that the two modes A and B "may correspond to spin flip transitions between the two QDs, or to spin transitions within the same QD". In the supplementary figure S2, they present a rather complex diagram with 5 transitions of various slopes. Can this spectroscopy be used to provide hints on the nature of the two modes A and B?*

Response: We really appreciate the Reviewer’s suggestion on using the spectroscopy to search for clues on the nature of the two modes A and B. We have re-examined our thinking and tried a variety of possibilities, and have now arrived at a two-hole model that has yielded an EDSR spectrum (figure 1 and S2) that mirrors our experimental observations. We have added the description in the last paragraph on page 3 in the revised main text:

“The observed resonances in Fig.1d are well described by a two-hole model built upon a single singlet in the (0, 2) charge configuration (S_{02}) and two-spin states $|\downarrow\uparrow\rangle, |\uparrow\downarrow\rangle, |\uparrow\uparrow\rangle$ and $|\downarrow\downarrow\rangle$ in the (1,1) charge configuration (see Supplementary Material, Sec. 2), as evidenced by Fig.1e and 1f. The large number of resonances and different slopes in Fig.1d hint at different g -factors for the two dots. Indeed, to generate the theoretical spectrum in Fig.1e, we use g -factors of 7 and 3.95 for the two dots. With such different g -factors, the two-spin states in the (1,1) regime would be the product states for any magnetic field above 0.1 T. In the following we focus on two resonances, denoted as mode A and mode B. The corresponding transitions involve single-spin-flip of the left (A, between $|\downarrow\downarrow\rangle$ and $|\downarrow\uparrow\rangle$) and the right (B, between $|\downarrow\downarrow\rangle$ and $|\uparrow\downarrow\rangle$) dot.”

New figure 1. e, Calculated EDSR spectrum from our effective two-hole model with g -factors of left and right dot of $g_L = 7$ and $g_R = 3.95$. The four highlighted resonances (purple, yellow, red and blue) correspond to spin-flips between $|\downarrow\downarrow\rangle$ and the other four eigenstates, indicated by the arrows in the energy level diagram f. Three related resonances are colored in grey in e (see details in Supplementary Material, Sec.2, EDSR spectrum).

To make our statements clear, we have added the discussion of the EDSR spectrum in the revised Supplementary Material, Sec. 2, as follows:

“We measure the EDSR signal by observing the variation in the leakage current in the PSB regime. When EDSR flips a spin, PSB is lifted and the current starts to flow, until another hole with the right spin state blocks the transport again. As such, EDSR leads to an increase in the leakage current. Notice that if none of the hole spins are flipped when driven, PSB cannot be lifted and leakage current would not change. Thus the change in PSB leakage current is a clear indication that a spin-flip transition has happened.

In our measurement, we apply a continuous microwave pulse to one of the gates (R). The electric field E_{ac} that drives EDSR can be estimated from Fig. S8c & d by modeling our device and inputting the a.c. voltage V_{ac} from the microwave source. Under different magnetic fields and microwave frequencies, more than 5 oblique lines with anti-crossings are obtained at a low microwave power $P = -15$ dBm (Fig. S2a) indicating that the microwave is on resonance with a particular spin-flip transition. In the spectroscopy, drifts in the transport current account for current variation along the longitudinal axis while the step-like signal along the horizontal axis is attributed to the attenuation difference of the circuit.

The most commonly observed Pauli Spin Blockade happens near the $(0,2) - (1,1)$ transition, while our DQD contains roughly five to ten holes in each dot. Nevertheless, we find that an effective two-hole model (for valence holes involved in the observed PSB) produces a spectrum that fits our observation quite well. In this model, we include basis states of as single-dot singlet, S_{02} , and four two-hole states in the $(1,1)$ regime: $|\downarrow\uparrow\rangle$, $|\uparrow\downarrow\rangle$, $|\uparrow\uparrow\rangle$ (T_+) and $|\downarrow\downarrow\rangle$ (T_-). Here, only the triplets in the $(1,1)$ regime are considered since T_{02} has much higher energy than S_{02} due to Pauli exclusion principle. The corresponding two-hole Hamiltonian can be written as

$$H_{DQD} =$$

$$\begin{pmatrix} -\varepsilon & t/\sqrt{2} & -t/\sqrt{2} & -\Delta_{SO}^* & \Delta_{SO} \\ t/\sqrt{2} & 1/2(g_1 - g_2)\mu_B B & 0 & 0 & 0 \\ -t/\sqrt{2} & 0 & -1/2(g_1 - g_2)\mu_B B & 0 & 0 \\ -\Delta_{SO} & 0 & 0 & -1/2(g_1 + g_2)\mu_B B & 0 \\ \Delta_{SO}^* & 0 & 0 & 0 & 1/2(g_1 + g_2)\mu_B B \end{pmatrix},$$

where g_1 and g_2 are the g -factors of the hole spin in the two dots respectively, t is the spin-independent tunnel coupling between the two dots, Δ_{SO} is the spin-flip tunnel coupling between $|\uparrow\uparrow\rangle / |\downarrow\downarrow\rangle$ and S_{02} , and ε is the detuning of $S_{11} = 1/\sqrt{2}(|\uparrow\downarrow\rangle + |\downarrow\uparrow\rangle)$ with respect to S_{02} . The calculated eigenvalues of the matrix are given in Fig. S2b,

showing the five eigenstates as a function of the magnetic field B . The spectral curves arising from the transitions between any two states can be mapped to this energy spectrum. Energetically, spin blockade should happen when the low-energy (1,1) triplet $|\downarrow\downarrow\rangle$ is occupied, and spin-flip transition from this state to any other would lead to an increase in current and thus an EDSR signal. The high-energy triplet $|\uparrow\uparrow\rangle$ is usually unoccupied, especially at higher magnetic field, thus transitions originating from it are generally not observed experimentally. On the other hand, at a very low field it does mix with S_{02} significantly by spin-flip tunneling, such that a seemingly two-spin-flip transition can be seen in the experimental measurement, though that particular signal quickly faded away as the magnetic field is increased.

When operating the spins in experiment, the working position is fixed at a large value of detuning (point M in Fig. 1b) deeply in the Coulomb blockade regime, instead of the low-detuning regime ($\epsilon \sim 0$) where the EDSR spectrum is measured. To clarify the manipulated spin states more clearly, we perform the spectral calculation in Fig. S2 c & d by assuming $\epsilon \sim 100 \mu\text{eV}$. The linear resonances can be achieved in this case where rotation of a single spin (either left or right) can be performed for coherent control (Fig. S2 c).”

Fig. S2 is modified for better understanding as below:

Original Figure S2: EDSR spectrum. A differential signal of the EDSR spectrum is obtained at microwave power $P = -15$ dBm. Qubit frequencies of both mode A and mode B are fixed around 8 GHz (circle and square), which are far from any anti-crossing.

Revised Figure S2: EDSR spectrum. **a.** Simulated and experimental results of the EDSR spectrum. During spin operation, qubit frequencies of both mode A (yellow) and mode B (purple) are fixed around 8 GHz, which are far from any anti-crossing. In the left panel of **a**, calculated seven transitions between $|T_{\downarrow}\rangle$ (four colored and three grey curves) and other four states are mapped as a function of magnetic field B . Four colored resonance are indicated by arrows in **b**. Eigenstates, as well as mixing, of these five states vary with the magnetic field. Parameters used for simulation are $g_L = 7$, $g_R = 3.95$, $2t = 14 \mu\text{eV}$, $\varepsilon = 1.5 \mu\text{eV}$ and $|\Delta_{SO}| = 3.8 \mu\text{eV}$. **c & d.** Simulated spectrum and eigenstates at a large detuning $\varepsilon = 100 \mu\text{eV}$. The coherent control of the left (right) spin is performed in the yellow (purple) resonance.

In particular, do the authors have an explanation for the important g -factor difference between these two modes (resp. 4.5 and 3).

Response: First of all, the previous g -factors of 4.5 and 3 were simply obtained from the slopes of the resonances near a splitting of 8 GHz. In the two-hole model we adopt now, the energy splittings are determined by not only Zeeman splitting but also tunnel coupling and exchange splitting, which result in the g -factors of 7 and 3.95 in the revised manuscript. Nevertheless, the g -factors that give best fitting to the experimental data still have a large difference between the two dots. Our understanding is that first, g -factor in a confined system is generally quite different from that in the bulk, and second, our two dots have a quite different occupation, so that the valence holes most probably occupy different orbital states, and different g -factors are to be expected.

We have thus added the following discussion on g -factor in the 2nd paragraph on page 5 of the revised Supplementary Material:

“The g -factor for a hole spin depends on control parameters such as the inter-dot tunnel coupling (Fig. S3). To verify that the Rabi frequency can be further increased, we have compared the Rabi frequency as a function of pulse height in Fig. S4, where a deeper working position leads to a smaller Rabi frequency. Here, we obtain the two g -factors as $g_L = 7$ and $g_R = 3.95$. For heavy holes in Ge hut wire, one expects a small in-plane g -factor of 0.2 and a large out-of-plane g -factor of 21.4^{S1}. In our model, we assume two different g -factors for the spins in the left dot and the right dot respectively. We believe this difference can be attributed to the unequal hole occupations between the two dots. Our observations seem to indicate that fewer holes occupy the left dot compared to the right dot (Fig. S1b & Fig. S8b). It is thus quite probable that the wave function of the manipulated spin differs in the two dots. Moreover, a recent preprint^{S2} shows a large out-of-plane g -factor of 15.7 and very different g -factors due to a different hole filling. In short, the g -factor difference between mode A and mode B is quite understandable considering their different occupations and states involved.”

We have also added a new Fig.S3, where we explored how the g -factor for mode B varies as we change the voltage on the middle gate M, which clearly illustrates how g -factor is also affected by the electric potential we apply on the gates around a quantum dot.

New Figure S3: EDSR of mode B at different gate voltages of gate M. The obtained g^* -factor ($\Delta E = g^* \mu_B B$) in the linear regime varies as a function of the gate voltage of middle gate M.

2. *Can the authors develop their explanation about the saturation of f_{Rabi} as P is increased above +9dBm? Would photon-assisted tunneling affect the spin manipulation or the readout?*

Response: To answer the question of saturation of Rabi frequency, we have added the following explanations in the 1st paragraph on page 9 of Supplementary Material.

“...In the case of a very strong microwave burst when P is above 9dBm in the same mode, the fast decay of Rabi oscillations (Fig. S6b) might be explained by PAT. A long microwave burst of strong microwave field helps the tunneling of a hole in either of the dots to reservoirs or between the dots by absorbing photons, which could result in the lift

of spin blockade. This process would lead to an increase of leakage to non-qubit states and accelerated decay of Rabi oscillations, which mainly affects spin manipulation.”

b

Figure S6b, Rabi Oscillations at higher microwave power. Rabi frequencies are obtained at 604 ± 2 , 644 ± 2 , 698 ± 2 MHz from bottom to top.

Is the T_2^ further reduced for $P=10, 11$ and 12 dBm?*

Response: We speculate that it would be further reduced at higher powers, although we are not equipped to measure these values in experiment. We have added an explanation and Fig. S6d in the 1st paragraph on page 9 of Supplementary Material.

“When the driving power P is increased to 10, 11 and 12 dBm, Rabi oscillation decays too fast for us to consistently generate the $\pi/2$ pulse with reasonable fidelity. We are thus unable to obtain T_2^* in these high-power cases. However, we did obtain T_2^* at lower powers (Fig. S6d) where T_2^* shows a downward trend as driving power increases, when $5 \text{ dBm} < P < 9 \text{ dBm}$. Based on these results, we believe that the value of T_2^* at $P=10, 11$ and 12 dBm would be further reduced.”

Revised Figure S6d, Dephasing time of mode A at different microwave power. $T_2^* = 66 \pm 6$ ns and $T_2^* = 42 \pm 4$ ns are extracted from Ramsey experiment at $P = 0$ dBm and $P = 9$ dBm respectively, $T_2^* = 65 \pm 2$ ns of mode B is measured at $P = 0$ dBm.

Could these limitations be mitigated by changing the manipulation position M or lowering the interdot tunnel-coupling during the MW application?

Response: This is a good question. We indeed believe that it is possible to further optimize our manipulation protocol and position. To address this question on optimization, we have added a new Fig.S4 in Supplementary Material and included the explanation in the 2nd paragraph on page 10 of the revised main text:

“...Moreover, a faster Rabi oscillation can be achieved by changing the inter-dot coupling or the manipulation position. Such optimization can be achieved by tuning the voltage of gate R. Our test results show that the Rabi frequency can be increased from 63 MHz to 111 MHz by switching the manipulation position from M2 to M1 (Fig. S4 in Supplementary Material). We are thus optimistic that a faster Rabi operation is achievable after optimization.”

New Figure S4: **a.** Schematic of the manipulation point in the stability diagram. **b.** Qubit frequency shifts as a function of pulse height. Each point is obtained by the chevron pattern of Rabi oscillations. The cases of M1 and M2 are shown in **c** and **d** with a different Rabi frequency. All the measurements in the main text are performed at the position of M2.

3. *I feel that the paper lacks perspective on a future improvement of this work. The authors mention that they “believe it is not the upper bound for the Rabi frequency of a GHW hole spin qubit”. Besides finding another mode with a stronger coupling to the MW drive, can the authors propose a strategy to further enhance the quality factor of their spin manipulation?*

Response: We thank the referee for this pertinent suggestion. To improve our manuscript in this respect, we have modified the section in the 2nd paragraph on page 10 of the revised main text:

“Even though the Rabi frequency of 540 MHz is limited by unwanted PAT and heating effects at high microwave power, we believe it is not the upper bound for the Rabi frequency of a GHW hole spin qubit. In principle faster operation is possible if another branch of EDSR with a larger Larmor frequency can be identified in the DQD. For example,

the transition between $|\downarrow\downarrow\rangle$ and S_{02} can also be observed in our system (red in Fig. 1e&1f). This spin-flip transition corresponds to a larger energy splitting compared to mode A and mode B at the same magnetic field, and could result in a larger Rabi frequency. Moreover, a faster Rabi oscillation can be achieved by changing the inter-dot coupling or the manipulation position. Such optimization can be achieved by tuning the voltage of gate R. Our test results show that the Rabi frequency can be increased from 63 MHz to 111 MHz by switching the manipulation position from M2 to M1 (Fig. S4 in Supplementary Material). We are thus optimistic that a faster Rabi operation is achievable after optimization.”

New Figure 1. e, Calculated EDSR spectrum from our effective two-hole model with g -factors of left and right dot of $g_L = 7$ and $g_R = 3.95$. The four highlighted resonances (purple, yellow, red and blue) correspond to spin-flips between $|\downarrow\downarrow\rangle$ and the other four eigenstates, indicated by the arrows in the energy level diagram f. Three related resonances are colored in grey in e (see details in Supplementary Material, Sec.2, EDSR spectrum).

4. *The paper by Froning et. al, discussed in the note at the end of the main text (ref [31]), should be added to the benchmark presented in Fig. 4 of the main text.*

Response: We thank the Reviewer’s suggestion. We have changed the Fig. 4b and included their results in the new Fig. S10 (old Fig.S9).

5. *In the supplementary material, the labels of the benchmark (Fig. S9) are incorrect, they should be shifted by 1.*

Response: We agree with the Reviewer’s comment, and have corrected the labels in Supplementary Material and revised relative captions.

Original Figure S9: Benchmarking spin qubit dephasing time and x -rotation speed for different qubits. Related works from the same group are not listed.

New and revised Figure S10: Qualitative benchmarking of dephasing time and x -rotation speed for different types of spin qubits. Related works from the same group are not listed.

While the main text presents a good English level and clear explanations, the supplementary material contains many mistakes and missing words, with some sentences hardly understandable at all. For example, the caption of Fig. S3 is extremely unclear.

Response: We apologize for the mistakes and missing words and inappropriate presentations in the Supplementary Material. We have used an academic English proofreading service (Springer Nature Author Services), all typos, grammatical errors and inaccurate statements have been corrected in the Revision.

Response to Reviewer 2

Comments: The manuscript “Ultrafast Operation of a Hole Spin Qubit in a Germanium Quantum Dot” by Wang et al. presents a thorough study of hole spin qubits in germanium hut-wires. The data convincingly show fast qubit manipulation, and clear extraction of parameters such as coherence times and Rabi frequencies. The interpretation of the data is very likely to be true, but I find the current presentation below standard for publication:

- 1. The manuscript does not give appropriate credit to other works in the field, esp fig.4b (=fig S9) must be adapted, since it has errors and omissions*
- 2. The statements about heavy / light hole behaviour are confusing*
- 3. The interesting EDSR pattern deserves more attention*
- 4. The nature of modes A and B is left vague*
- 5. Ref [31] must get the credit it deserves.*

The suggested revisions are below in my comments.

Response: We appreciate the Reviewer’s insightful and constructive suggestions for our work. The comment on the interpretation of the EDSR spectra echoes similar comments by Reviewer 1, making it clear that we did not do a good job in presenting our results in a credible picture. Therefore, we have spent some time examining different possibilities, and have developed an effective two-hole model as mentioned in the response to Reviewer 1. We have also made changes in the text in response to the recommendations and criticisms the reviewer has raised, as we describe in detail below.

MAJOR COMMENTS

- 1. Fig 4b / S9 are in principle a very nice overview to compare this work to results from literature. However, there are 3 major concerns:*
 - a. Is this a relevant way to plot it? If you want to plot the operation speed versus coherence, then the relevant coherence is the T_2 (instead of T_2^*). If you are interested the number of operations before decoherence, then that would be a more suitable benchmark. Even better would be the fidelity via randomized benchmarking, as in nearly all references.*

Response: We agree with the referee that randomized benchmarking is in general a better way to characterize gate fidelity, which we intend to pursue in the future. However, in this manuscript our point is to emphasize that the ultrafast operation is demonstrated without any echo pulses. As such it seems quite reasonable to us to compare Rabi frequency with T_2^* instead of T_2 , as pointed out by the referee. Therefore, we have modified Fig 4b and left Fig S10 unchanged.

Original Figure 4: Speed of x -rotation and results compared to the literatures. a, Linear dependence of Rabi frequency on the amplitude of applied driving field (i.e. E_{ac}). Rescale of power is converted to amplitude by P (mW) = $10^{(P \text{ (dBm)} - 36)/10}$, $P^{1/2} = (2 * P \text{ (mW)} * 50 \Omega)^{0.5}$. For $P > 9$ dBm, the data deviate from the linear dependence as a result of PAT effect (Supplementary Material, Fig. S5c). Parameters for mode A are $l_{\text{dot}} = 10$ nm, $\hbar\omega_y = 1$ meV, $g = 4.5$, $B = 100$ mT; Parameters for mode B are $l_{\text{dot}} = 10$ nm, $\hbar\omega_y = 1$ meV, $g = 3$, $B = 156$ mT. Spin-orbit length of mode A and B are obtained to be 2.3 nm and 4.5 nm respectively, by fitting the Rabi frequency's dependence on the square root of power using Eq. (2). **b,** Dephasing time and x -rotation speed (i.e. Rabi frequency for single spin qubit) of different spin qubits. Stars label shows our result of hole spin qubit with $T_2^* = 42$ ns and $f_{\text{Rabi}} = 540$ MHz. Light pink background marks the regime where $2f_x T_2^* \leq 100$. The references are listed in Supplementary Material.

Revised Figure 4: Rabi frequency and spin-orbit length. **a**, Linear dependence of Rabi frequency on the amplitude of applied driving field (i.e. E_{ac}). Rescale of power is converted to amplitude by P (mW) = $10^{(P \text{ (dBm)} - 36)/10}$, $P^{1/2} = (2 * P \text{ (mW)} * 50 \Omega)^{0.5}$. For $P > 9$ dBm, the data deviate from the linear dependence as a result of PAT effect (Supplementary Material, Fig. S6c). Parameters for mode A are $l_{\text{dot}} = 5$ nm, $\hbar\omega_y = 3$ meV, $g = 7$, $B = 100$ mT; Parameters for mode B are $l_{\text{dot}} = 5$ nm, $\hbar\omega_y = 3$ meV, $g = 3.95$, $B = 156$ mT. Spin-orbit length of mode A and B are obtained to be 1.5 nm and 1.4 nm respectively, by fitting the Rabi frequency's dependence on the square root of power using Eq. (2). **b**, Rabi oscillations of mode A and B at microwave power $P = 9$ dBm after subtracting the linear background.

b. Is this a fair way to plot it? Are all T_2^ values measured at the corresponding Rabi frequency? This does not seem to be the case for the data point of this work, while it does seem to hold for some of the literature values. That would be an unfair comparison.*

Response: This is an insightful question. We have obtained the fastest result of 540 MHz at a power of 9dBm and use the data point of $T_2^* = 42$ ns at the same power though dephasing time at other applied microwave powers are experimentally obtained as well (Fig. S6d). To make out if it is a fair comparison, we dig in the measurement details of each work and the table below shows the results.

	$f_{\text{Rabi}}/\text{MHz}$	Power of rf for Rabi/dBm	T_2^*	Power of rf for T_2^*/dBm
S2	0.5	N/A	270 μs	N/A
S3	0.3	5	120 μs	-20
S4	0.33/0.4	N/A	61 μs	N/A
S5	16.6	N/A	20 μs	N/A
S6	1	N/A	6.4 μs	N/A
S7	4	N/A	1.8 μs	N/A
S8	4.8	N/A	10 μs	N/A
S9	14	N/A	0.2 μs	N/A
S10	35	N/A	1.84 μs	N/A
S11	57	N/A	1.3 μs	N/A
S12	2	N/A	1 μs	N/A
S13	5	18.8	1 μs	18.4
S14	100	N/A	817 ns	N/A
S15	3	10	55 ns	10
S16	85	6.3	60 ns	8
S17	140	17.6	130 ns	11
S18	435	34	11 ns	3

The microwave power is not mentioned in most of the references. We believe this variable can indeed lead to different dephasing time, which is also observed in our experiment (fig. 3d & 3e) but it should not vary by orders of magnitude at a limited microwave power. Therefore, the listed T_2^* in Fig. S10 is just a qualitative comparison.

c. Important values are missing (ref S17, and ref 31), and the Rabi frequency of S14 is placed below 10 MHz while they reach 100 MHz. Since there are some errors, I would suggest to double check all values by two separate authors.

Response: We are really thankful to the referee for spotting these errors. Sorry for incorrectly linking the references and data points in the previous version. All the values are checked and errors are corrected (for example, old S17 and S14 are shown as S19 and S16 respectively in the new figure). A new reference ref S20 is added (i.e. old ref 31).

Original Figure S9: Benchmarking spin qubit dephasing time and x -rotation speed for different qubits. Related works from the same group are not listed.

New and revised Figure S10: Qualitative benchmarking of dephasing time and x -rotation speed for different types of spin qubits. Related works from the same group are not listed.

2. The current version of the manuscript is unclear about the nature of the holes: whether they are heavy or light holes, or a mix. The different statements are spread through the manuscript and are confusing:

p1: “the strong SOI of heavy holes in our GHW”

p2: “..the lowest-energy levels are nearly pure heavy holes (HHs) with a light hole (LHs) admixture below 1%, although larger dot occupation could increase this admixture. Due to the hole spin-3/2, the multiplicity of available HH and LH spin states leads to a complex pattern of spin transitions,”

p3: “..the uncertainty in the hole occupation number and the possibility of mixing between LH and HH states, ..”

p4: “Because of the relatively small number of holes per dot (~10) we assume EDSR is mediated by the strong SOI of 2D heavy-holes.”

p5: “ m^* the Ge HH effective mass,”

p6: “The spin-orbit length difference between modes A and B comes from different states involved in the respective transitions,”

a. I would advise to combine these in 1 paragraph, and/or add a discussion at the end of the manuscript about what can/cannot be concluded/speculated. HH-LH physics is a hot topic in our field, so a concise discussion will add value to the manuscript.

Response: We agree with the reviewer’s comment on the importance of the nature of the holes. We have revised p2, p3, p4 and p6 in the following and added Section 9 in Supplementary Material to discuss the heavy hole physics in our system.

Revised main text of p2: ~~“Due to the hole spin 3/2, the multiplicity of available HH and LH spin states leads to a complex pattern of spin transitions, in particular in the case of two spin states, suggesting different spin-orbit coupling parameters in these transitions.”~~

Revised main text of p3: ~~“...the uncertainty in the hole occupation number and the possibility of mixing between LH and HH states, ...”~~

Revised main text of p4: ~~“Because of the relatively small number of holes per dot (~10) we assume EDSR is mediated by the strong SOI of 2D heavy-holes. With a three dimensional confinement and an out-of-plane magnetic field, the lowest states in the subspace spanned by the spin-3/2 hole states can be calculated, yielding states that are 98%~~

HH (see Supplementary Material, Sec. 9). With the relatively small number of holes per dot (5 to 10), we attribute the manipulated spin states to be HH throughout the manuscript.”

Revised main text of p6: ~~“The spin-orbit length difference between modes A and B comes from different states involved in the respective transitions”~~

Added Supplementary Material, Sec. 9: **“Section 9. Heavy hole states in GHW** Considering the HH and LH bands of Ge and assuming that the HW is free of shear strain, the Hamiltonian for a two-dimensional quantum-dot-confined hole in the presence of a magnetic field is^{S21}

$$H = \frac{\hbar}{2m} \left[\left(\gamma_1 + \frac{5\gamma_2}{2} \right) k^2 - 2\gamma_2 \sum_v k_v^2 J_v^2 - 4\gamma_3 (\{k_x, k_y\} \cdot \{J_x, J_y\} + \text{c. p.}) \right] + 2\mu_B \mathbf{B} \cdot (\kappa \mathbf{J} + \mathbf{qJ}^3) + b \sum_v \epsilon_{vv} J_v^2 + V(x, z).$$

It consists of the Luttinger-Kohn Hamiltonian, the Bir-Pikus Hamiltonian and the confinement in the transverse directions $V(x, z)$. We follow the calculation in References S22 and S23, and treat our potential as a rectangle hard-wall potential of width L_x and height L_z for simplicity, i.e. $V(x, z) = 0$ if $|x| < \frac{L_x}{2}$ and $|z| < \frac{L_z}{2}$ and $V(x, z) = \infty$ otherwise^{S22}, where the three axes x, y, z are oriented along the width, length and height, respectively, of the HW. The applied magnetic field is $\mathbf{B} = (B_x, B_y, B_z)$, and the kinetic momentum is $\hbar \mathbf{k} = -i\hbar/\nabla + e\mathbf{A}$ where $\mathbf{B} = \nabla \times \mathbf{A}$. The vector potential is chosen as $\mathbf{A} = (B_y z - B_z y, -\frac{B_x z}{2}, \frac{B_x y}{2})$ for convenience. We use the basis of

$$|j_z, n_x, n_y, n_z\rangle = |j_z\rangle \otimes |\varphi_{n_x, n_y, n_z}\rangle$$

with the envelope function

$$\varphi_{n_x, n_y, n_z}(x, y, z) = \frac{2}{\sqrt{L_x L_z}} \sin \left[n_x \pi \left(\frac{x}{L_x} + \frac{1}{2} \right) \right] \times \sin \left[n_y \pi \left(\frac{y}{L_y} + \frac{1}{2} \right) \right] \times \sin \left[n_z \pi \left(\frac{z}{L_z} + \frac{1}{2} \right) \right].$$

We then project the Hamiltonian to a subspace with $n_x \leq 3$, $n_y \leq 3$ and $n_z \leq 3$, using the band structure parameters of bulk Ge $\gamma_1 = 13.35, \gamma_2 = 4.25, \gamma_3 = 5.69, \kappa = 3.41, q = 0.07$. The values for the strain tensor elements are $\epsilon_{xx} = \epsilon_{yy} = -0.033$ and $\epsilon_{zz} = 0.02$ ^{S24}.

Using a magnetic field of 100 mT (close to mode A and B) along the z direction as in the experiment, $L_x = 20$ nm, $L_y = 57$ nm (inset of Fig. S9b), $L_z \leq 2$ nm and other parameters, we diagonalize the (108×108) matrix of the Hamiltonian, and obtain a spin

expectation value of $\langle J_z \rangle \approx 1.48$, from which we find the probability of HH to be $p_{\text{HH}} \approx 98\%$ from $\frac{3}{2}p_{\text{HH}} + \frac{1}{2}(1 - p_{\text{HH}}) \approx 1.48$. Clearly, this is a nearly pure heavy hole system.

Moreover, a large LH and HH splitting of $\frac{2\gamma_2\hbar^2\pi^2}{mL_z^2} \geq 1.6 \text{ eV}$ is obtained with $L_z \leq 2 \text{ nm}$ according to ref.S22, making it unlikely to have significant LH component for any low-energy hole states.

We have also considered the case when an in-plane magnetic field is applied. Similar to the case above, we find a 98% HH probability when $B = 0.1\text{T}$. Therefore, we can conclude that the holes in our system are close to pure heavy holes no matter what the applied magnetic field direction is.”

b. The estimate of the hole number should be substantiated with data and arguments. Is it similar for the two quantum dots?

Response: We thank the referee for this request of clarity, and have added the charge stability diagram as the new Fig. S1b. We can conclude that the number of holes is relative small per dot even if there may still additional holes due to the limitation of small tunneling rate corresponding to the inset of Fig. S8b. It seems to us that there are different occupation numbers in the two dots. Around two holes occupy the left dot, while at least five holes can be detected in the right dot. We believe it is safe to state that the occupation number of the left dot is smaller than that of the right dot.

Original Figure S1: Schematic representation of the device and the positive bias triangle. **a**, Schematic representation of the three-gate device. The GHW, which consists of a Si cap and a Ge layer, is grown on Si substrate and covered by a layer of aluminum oxide. Three gates are deposited on top of this insulating layer and HW. **b**, A conductance

bias triangle of the DQD at $V_{sd} = -3$ mV, which can be compared to Fig. 1b in the main text (a conductance bias triangle at $V_{sd} = 3$ mV).

Revised Figure S1: Schematic representation of the device and the stability diagram.

a, Schematic representation of the three-gate device. The GHW, which consists of a Si cap and a Ge layer, is grown on Si substrate and covered by a layer of aluminum oxide. Three gates are deposited on top of this insulating layer and HW. **b**, Stability diagram of the double quantum device at $V_{sd} = 2$ mV. The dashed rectangle is where we measure the conductance triangle. An intra-dot Coulomb energy, i.e. orbital splitting, of 10 meV is extracted from ΔV_{R1} . **c**, The conductance bias triangle of the DQD at $V_{sd} = 3$ mV (the same as Fig. 1b) and the inter-dot Coulomb interaction of 0.5 meV obtained from ΔV_{R2} . **d**, The conductance bias triangle at the same regime with of a bias of $V_{sd} = -3$ mV, a large leakage current can be observed at $\epsilon \sim 0$.

Revised Figure S8b, Simulated z component of static electric field E_{ac}^z after applying voltages of $V_L = 0.17$ V, $V_M = 0.085$ V, $V_R = 0.355$ V and $V_{sd} = 3$ mV. Inset: E_{ac}^z as a function of position along y direction (linecut along the dash line). A potential width of 57 nm is obtained after Gaussian fitting.

3. *The complex pattern of EDSR transitions in figure S2 is very interesting and should be explained clearer. This a great opportunity to gain a deeper understanding of HH and LH spins.*
 - a. *At least a speculation of the origin of the EDSR lines and anti-crossings should be added to the manuscript.*

Response: As mentioned in the response to Reviewer 1’s first question, we have developed an effective two-hole model to describe the complex pattern of EDSR transitions, and the theoretical results match the experimental observed spectrum quite well. To make our statements clear, we have revised the discussion of EDSR spectrum in the revised Supplementary Material, Sec. 2, as follows:

“We measure the EDSR signal by observing the variation in the leakage current in the PSB regime. When EDSR flips a spin, PSB is lifted and the current starts to flow, until another hole with the right spin state blocks the transport again. As such, EDSR leads to an increase in the leakage current. Notice that if none of the hole spins are flipped when driven, PSB cannot be lifted and leakage current would not change. Thus the change in PSB leakage current is a clear indication that a spin-flip transition has happened.

In our measurement, we apply a continuous microwave pulse to one of the gates (R). The electric field E_{ac} that drives EDSR can be estimated from Fig. S8c & d by modeling our device and inputting the a.c. voltage V_{ac} from the microwave source. Under different

magnetic fields and microwave frequencies, more than 5 oblique lines with anti-crossings are obtained at a low microwave power $P = -15$ dBm (Fig. S2a) indicating that the microwave is on resonance with a particular spin-flip transition. In the spectroscopy, drifts in the transport current account for current variation along the longitudinal axis while the step-like signal along the horizontal axis is attributed to the attenuation difference of the circuit.

The most commonly observed Pauli Spin Blockade happens near the $(0,2) - (1,1)$ transition, while our DQD contains roughly five to ten holes in each dot. Nevertheless, we find that an effective two-hole model (for valence holes involved in the observed PSB) produces a spectrum that fits our observation quite well. In this model, we include basis states of as single-dot singlet, S_{02} , and four two-hole states in the $(1,1)$ regime: $|\downarrow\uparrow\rangle$, $|\uparrow\downarrow\rangle$, $|\uparrow\uparrow\rangle$ (T_+) and $|\downarrow\downarrow\rangle$ (T_-). Here, only the triplets in the $(1,1)$ regime are considered since T_{02} has much higher energy than S_{02} due to Pauli exclusion principle. The corresponding two-hole Hamiltonian can be written as

$H_{DQD} =$

$$\begin{pmatrix} -\varepsilon & t/\sqrt{2} & -t/\sqrt{2} & -\Delta_{SO}^* & \Delta_{SO} \\ t/\sqrt{2} & 1/2(g_1 - g_2)\mu_B B & 0 & 0 & 0 \\ -t/\sqrt{2} & 0 & -1/2(g_1 - g_2)\mu_B B & 0 & 0 \\ -\Delta_{SO} & 0 & 0 & -1/2(g_1 + g_2)\mu_B B & 0 \\ \Delta_{SO}^* & 0 & 0 & 0 & 1/2(g_1 + g_2)\mu_B B \end{pmatrix},$$

where g_1 and g_2 are the g-factors of the hole spin in the two dots respectively, t is the spin-independent tunnel coupling between the two dots, Δ_{SO} is the spin-flip tunnel coupling between $|\uparrow\uparrow\rangle / |\downarrow\downarrow\rangle$ and S_{02} , and ε is the detuning of $S_{11} = 1/\sqrt{2}(|\uparrow\downarrow\rangle + |\downarrow\uparrow\rangle)$ with respect to S_{02} . The calculated eigenvalues of the matrix are given in Fig. S2b, showing the five eigenstates as a function of the magnetic field B . The spectral curves arising from the transitions between any two states can be mapped to this energy spectrum. Energetically, spin blockade should happen when the low-energy $(1,1)$ triplet $|\downarrow\downarrow\rangle$ is occupied, and spin-flip transition from this state to any other would lead to an increase in current and thus an EDSR signal. The high-energy triplet $|\uparrow\uparrow\rangle$ is usually unoccupied, especially at higher magnetic field, thus transitions originating from it are generally not observed experimentally. On the other hand, at a very low field it does mix with S_{02} significantly by spin-flip tunneling, such that a seemingly two-spin-flip transition can be seen in the experimental measurement, though that particular signal quickly faded away as the magnetic field is increased.

When operating the spins in experiment, the working position is fixed at a large value of detuning (point M in Fig. 1b) deeply in the Coulomb blockade regime, instead of the low-detuning regime ($\epsilon \sim 0$) where the EDSR spectrum is measured. To clarify the manipulated spin states more clearly, we perform the spectral calculation in Fig. S2 c & d by assuming $\epsilon \sim 100 \mu\text{eV}$. The linear resonances can be achieved in this case where rotation of a single spin (either left or right) can be performed for coherent control (Fig. S2 c).”

Revised Figure S2: EDSR spectrum. a. Simulated and experimental results of the EDSR spectrum. During spin operation, qubit frequencies of both mode A (yellow) and mode B (purple) are fixed around 8 GHz, which are far from any anti-crossing. In the left panel of a, calculated seven transitions between $|\downarrow\rangle$ (four colored and three grey curves) and other four states are mapped as a function of magnetic field B . Four colored resonance are indicated by arrows in b. Eigenstates, as well as mixing, of these five states vary with the

magnetic field. Parameters used for simulation are $g_L = 7$, $g_R = 3.95$, $2t = 14 \mu\text{eV}$, $\varepsilon = 1.5 \mu\text{eV}$ and $|\Delta_{\text{SO}}| = 3.8 \mu\text{eV}$. **c & d.** Simulated spectrum and eigenstates at a large detuning $\varepsilon = 100 \mu\text{eV}$. The coherent control of the left (right) spin is performed in the yellow (purple) resonance.

b. The authors assign g -factors of 4.5 and 3 to mode A and B. I cannot retrieve where in the manuscript they write how they found these numbers. It looks like they used the f and B where the measurements were performed (~ 8 GHz). I am not sure whether this is justified, because that would imply that the g -factor strongly changes as the magnetic field is reduced to the anti-crossings (~ 2 -3 GHz), e.g. mode A has $f \sim 3$ GHz at $B \sim 20$ mT, corresponding to $g \sim 10$. Following the same argument, the mode with $f \sim 6$ GHz at $B \sim 10$ mT would have $g \sim 43$. I would expect some discussion/explanation in the manuscript about the g -factors: Why these values? Why so different for two similar QDs?

Response: As mentioned in the response to the first question of Reviewer 1, the previous g -factors of 4.5 and 3 are obtained in the linear regime of EDSR, away from the anti-crossings, by attributing the slope there purely to Zeeman splitting. With the new two-hole model, which fit our experiment very well, we have added discussions of g -factor in the last paragraph on page 3 of the main text and the 2nd paragraph on page 5 of the Supplementary Material.

Revised main text: “The observed resonances in Fig.1d are well described by a two-hole model built upon a single singlet in the (0, 2) charge configuration (S_{02}) and two-spin states $|\downarrow\uparrow\rangle, |\uparrow\downarrow\rangle, |\uparrow\uparrow\rangle$ and $|\downarrow\downarrow\rangle$ in the (1,1) charge configuration (see Supplementary Material, Sec. 2), as evidenced by Fig.1e and 1f. The large number of resonances and different slopes in Fig.1d hint at different g -factors for the two dots. Indeed, to generate the theoretical spectrum in Fig.1e, we use g -factors of 7 and 3.95 for the two dots. With such different g -factors, the two-spin states in the (1,1) regime would be the product states for any magnetic field above 0.1 T. In the following we focus on two resonances, denoted as mode A and mode B. The corresponding transitions involve single-spin-flip of the left (A, between $|\downarrow\downarrow\rangle$ and $|\uparrow\downarrow\rangle$) and the right (B, between $|\downarrow\downarrow\rangle$ and $|\uparrow\downarrow\rangle$) dot.”

New Figure 1. e, Calculated EDSR spectrum from our effective two-hole model with g -factors of left and right dot of $g_L = 7$ and $g_R = 3.95$. The four highlighted resonances (purple, yellow, red and blue) correspond to spin-flips between $|\downarrow\downarrow\rangle$ and the other four eigenstates, indicated by the arrows in the energy level diagram **f**. Three related resonances are colored in grey in **e** (see details in Supplementary Material, Sec.2, EDSR spectrum).

Revised Supplementary Material: “Here, we obtain the two g -factors as $g_L = 7$ and $g_R = 3.95$. For heavy holes in Ge hut wire, one expects a small in-plane g -factor of 0.2 and a large out-of-plane g -factor of 21.4^{S1}. In our model, we assume two different g -factors for the spins in the left dot and the right dot respectively. We believe this difference can be attributed to the unequal hole occupations between the two dots. Our observations seem to indicate that fewer holes occupy the left dot compared to the right dot (Fig. S1b & Fig. S8b). It is thus quite probable that the wave function of the manipulated spin differs in the two dots. Moreover, a recent preprint^{S2} shows a large out-of-plane g -factor of 15.7 and very different g -factors due to a different hole filling. In short, the g -factor difference between mode A and mode B is quite understandable considering their different occupations and states involved.”

Why do they change strongly with B? how are the values coupled to the picture of HHs, LHs and possible mixing?

Response: We do not think g -factor should change with magnetic field in our EDSR spectrum. It’s our mistake to simply obtain the g -factors in the linear regime in the previous version. In the current manuscript, we obtain the two g -factors as $g_L = 7$ and $g_R = 3.95$ based on our effective two-hole model. Furthermore, g -factors depend on other parameters in our system, such as the hole occupancy and the related orbital states.

We have thus added “The g -factor for a hole spin depends on control parameters such as the inter-dot tunnel coupling (Fig. S3). To verify that the Rabi frequency can be further increased, we have compared the Rabi frequency as a function of pulse height in Fig. S4, where a deeper working position leads to a smaller Rabi frequency. Here, we obtain the two g -factors as $g_L = 7$ and $g_R = 3.95$” in the 2nd paragraph on page 5 of Supplementary Material.

- 4. In comments 2 and 3 above, I understand that many questions regarding g -factors and HH-LH cannot be answered based on the data. This is exactly why it is so important to at least clearly state the factual observations, what can be concluded and discuss possible explanations / speculations for what cannot be concluded. A clear distinction between what can and cannot be concluded can help the field to follow-up with new theory and experiments.*

Response: We agree with the reviewer that some questions remain unsolved regarding the complex EDSR pattern, even though our new model actually fits our experiment quite well. In the revised manuscript, we can conclude that the the EDSR spectrum can be explained by our two-hole model. Additionally, the calculation result of heavy hole states (Sec. 9 in Supplementary Material) convince us that HH dominates the system. Of course, there are still some problem remains to be studied. For example, we cannot clarify whether excited states contribute to theses resonances in any way. Besides, we are now trying to further develop our model for our future work to better understand the rich physics in our EDSR spectrum, such as the odd-even effect that has been studied in InAs nanowire (*Phys. Rev. Lett.* 112, 227601 (2014)).

- 5. Page 5: “The spin-orbit length difference between modes A and B comes from different states involved in the respective transitions,”. Do modes A and B correspond to the left and right quantum dot? If not, what could explain the modes (speculation is allowed)? Adding plot of T_2^* for mode A and mode B versus a useful parameter (e.g. f or f_{Rabi} or B or power) can help the reader. The origin of the differences between mode A and B must be discussed in the manuscript, similar to the 2 dots, perhaps in same paragraph.*

Response: The referee has the right intuition here. Based on our model, modes A and B indeed correspond to the left and right spin in the two dots as discussed above. To make the manuscript more informative, we have modified Fig. S6d to compare the differences of T_2^* .

Revised Figure S6d, Dephasing time of mode A at different microwave power. $T_2^* = 66 \pm 6$ ns and $T_2^* = 42 \pm 4$ ns are extracted from Ramsey experiment at $P = 0$ dBm and $P = 9$ dBm respectively, $T_2^* = 65 \pm 2$ ns of mode B is measured at $P = 0$ dBm.

To discuss the origin of the differences between mode A and B, we include the section in the 2nd paragraph on page 5 of the revised Supplementary Material as follows:

“The g -factor for a hole spin depends on control parameters such as the inter-dot tunnel coupling (Fig. S3). To verify that the Rabi frequency can be further increased, we have compared the Rabi frequency as a function of pulse height in Fig. S4, where a deeper working position leads to a smaller Rabi frequency. Here, we obtain the two g -factors as $g_L = 7$ and $g_R = 3.95$. For heavy holes in Ge hut wire, one expects a small in-plane g -factor of 0.2 and large out-of-plane g -factor of 21.4^{S1}. In our model, we assume two different g -factors for the spins in the left dot and the right dot respectively. We believe this difference can be attributed to the unequal hole occupations between the two dots. Our observations seem to indicate that fewer holes occupy the left dot compared to the right dot (Fig. S1b & Fig. S8b). It is thus quite probable that the wave function of the manipulated spin differs in the two dots. Moreover, a recent preprint^{S2} shows a large out-of-plane g -factor of 15.7 and very different g -factors due to a different hole filling. In short, the g -

factor difference between mode A and mode B is quite understandable considering their different occupations and states involved.”

6. Page 7: “While writing this letter, we became aware of a related preprint in which ultrafast control of hole spins is achieved in Ge/Si core/shell nanowire[31], which is quite different from our work”. Reference 31 may have been noticed during writing, but is so close to this work that it cannot be mentioned only as a final note. I agree that in details the work is different, but a journal like Nature Communications deserves the big picture to be painted. The works are too similar to ignore in the main text. The similarities are (i) the Ge/Si material system (ii) the ultrafast operation (540 MHz versus 435 MHz Rabi) and (iii) short spin-orbit length (2.3 nm versus 3 nm). This manuscript still has the record, and must be published with due credit for ref [13], meaning not in an added note.

Response: We agree with the Referee on properly citing the recent work. We have added a description in the introduction to give the work (ref [31]) appropriate credit in 2nd paragraph on page 2 and last paragraph on page 9 in the revised main text.

“A recent experiment in Ge/Si core/shell nanowire has reached a Rabi frequency of 435 MHz with a spin-orbit length of 3 nm⁴¹.”

“Ultrafast control of hole spins has also been achieved in a Ge/Si core/shell nanowire⁴¹, though the hole is quite different from ours. The key difference stems from geometry. Our Ge hut wire has a confinement potential closer to that of a quantum well, for which theory predicts a spin-3/2 (i.e. ‘heavy hole’) ground state. Ge/Si core/shell nanowires have cylindrical symmetry, where theory predicts a spin-1/2 (‘light-hole’) ground state. Due to possible mass reversal, the magnitude of the effective mass is not an indicator of spin character. On the other hand, the two system does share the feature of a strong spin-orbit coupling, which enables the fast control.”

MINOR COMMENTS

7. I do not understand the difference between the top plot in figure 2d (Rabi frequency of 542 MHz at $P = 9\text{dBm}$ and $f = 7.92\text{ GHz}$) and figure 3e (Rabi frequency of $\sim 100\text{ MHz}$ (my estimate) at $P = 9\text{dBm}$ and $f = 7.92\text{ GHz}$ (?)). Figure 4a implies that a Rabi frequency of $\sim 100\text{ MHz}$ is found at $P \sim -5\text{dBm}$.

Response: The result in Fig. 3e corresponds to Ramsey oscillations rather than Rabi oscillations. We apologize for not clarifying this clearly in the original manuscript. We have modified the caption to make this point more clear.

Figure 3: Free evolution and dephasing of the qubit. **a**, Ramsey fringes measured via transport current for mode A as functions of the driving microwave frequency detuning Δf and free evolution time τ between two $\pi/2$ pulses at microwave power $P = -10$ dBm, Larmor frequency $f = 7.92$ GHz and magnetic field $B = 100$ mT, where the second pulse have the form of $E_{ac} \cos(2\pi f + \Delta\phi)$. **b**, Frequency FFT corresponding to the Ramsey fringes in **a**. Two white dash lines mark the dependence of z -axis rotation on the frequency detuning Δf . When $\Delta f = 0$, $f_{\text{Ramsey}} = 0$ as well, corresponding to a pure dephasing process. **c**, Oscillations of the phase control of the second pulse at $\tau = 0$ ns and $\Delta f = 0$ MHz shows a perfect cycle of 2π . When performing Ramsey experiment at the condition of $\Delta\phi = 0, \frac{\pi}{2}, \pi$ and $\frac{3\pi}{2}$, the second pulse induces a rotation around the $x, y, -x$ and $-y$ axis, respectively. **d & e**, Dephasing times for mode A at $T_2^* = 84 \pm 9$ ns and $T_2^* = 42 \pm 4$ ns are obtained from the decay of ΔI extracted from Ramsey experiment with a fixed Δf by fitting $I = A \cdot \cos(f\tau + \phi_0) \exp(-(\tau/T_2^*)^2) + I_0$ at $P = -10$ dBm and $P = 9$ dBm, respectively. **f**, Pulse sequence used to control the free evolution of the qubit (top). Dephasing time for mode B: $T_2^* = 65 \pm 2$ ns at $P = 0$ dBm,

$f = 7.88$ GHz and $B = 156$ mT. **g**, Hahn echo sequence (top). After decoupling, dephasing time of the mode B qubit has been extended up to 523 ns. The durations of the pulses are $t_{\pi/2} = 2.5$ ns and $t_{\pi} = 5$ ns. Here we use $I = A * \exp(-(\frac{\tau}{T_2^{\text{Hahn}}})^{1+\alpha})$ to fit our data, where $\alpha = 0.9$ is determined by the noise spectrum. (see Supplementary Material, Fig. S5).

8. *Figure 1:*

a. *I find the dashed lines in fig 1b dangerously suggestive and would prefer the data without the dashed lines, or less pronounced. The reader should be able to judge by him/herself, which is not possible in the current version with thick overlaid lines.*

Response: The artificial line at detuning $\varepsilon \sim 0$ has been removed and the other thick overlaid lines have been replaced by thinner lines for clarity.

b. *The caption of 1b refers to a conductance triangle, which is absent in the data.*

Response: We have marked the triangle contour by the blue dashed lines connecting different current peaks in new figure 1b.

c. *The caption of 1b refers to “a suppressed current of ~ 1 pA at detuning ~ 0 , highlighted by the base of the triangle”. The detuning (axis) is not explained/defined, and the ~ 1 pA is not visible, at least nothing stands out from the background coulomb blockade. A line trace may help.*

Response: Thanks for the suggestion. We have made modifications to the figure and its caption, including adding an inset.

Original Figure 1b. A conductance triangle of the DQD at $V_{sd} = 3$ mV. A suppressed current of ~ 1 pA is observed at detuning ~ 0 , highlighted by the base of the triangle. Due to the weak signal of the small current, we mark the transitions by blue dash lines. The points R, I and M mark the readout, initialization and manipulation positions respectively.

Revised Figure 1b, A conductance triangle of the DQD at $V_{sd} = 3$ mV. Due to the weak signal of the small current, we mark the transitions of the triangle by blue dashed lines while the arrow indicates the position of the detuning $\varepsilon = 0$ (the energy difference between the left dot and the right dot) at the base line of the triangle. A suppressed current of ~ 1 pA is observed, which can be lifted by spin resonance. Points R, I and M mark the readout, initialization and manipulation positions respectively. Inset: a line trace at $V_L = 0.215$ V.

d. Fig 1c does not explain EDSR adequately.

Response: We agree this figure is inadequate and thus add “As described in our model, Δr makes contribution to the energy shift mediated by the spin-orbit coupling Δ_{SO} .” in the caption of revised Fig. 1c. We hope this figure would be helpful for readers to understand the mechanism of EDSR in a simple physical picture.

Figure 1c, Schematic of spin-orbit-coupling-mediated spin flip: the microwave electric field generated by the gate creates oscillatory displacements of the hole wave function (Δr) and its energy. As described in our model, Δr makes contribution to the energy shift mediated by the spin-orbit coupling Δ_{SO} . Such orbital dynamics lead to spin-flip with the help of SOI.

e. The caption of 1d refers to “a continuous microwave with a power of -15 dBm.” Perhaps the authors can add that this is applied at point R/I in fig 1b?

Response: Thanks for spotting the oversight. Modifications have been made in figure 1 and colored captions are added.

Figure 1d, EDSR spectrum, measured by applying a continuous microwave with a power of -15 dBm at the point R/I. The circle and square symbols show the working points for subsequent experiments, corresponding to a Larmor frequency ~ 8 GHz.

9. Figure 2a:

- a. It may help to add the positions M and R/I above the left and right panels
- b. I find the 2 cartoons in R/I unclear. Sequential or 2 different situations? Should be made clearer.

Response: Thanks for the suggestions. We have made some clarifying modifications in Figure 2a.

Original Figure 2a

Revised Figure 2a

10. Figure 3f-g: the number of significant digits in 65 ± 2 ns in 3f is different from the number in 523 ± 41 ns in 3g. Is this justified?

Response: The difference of error in Figure 3 comes from the fitting. It could be estimated that a smaller value of 65ns would lead to a smaller error with almost the same condition of convergence.

Response to Reviewer 3

Comments: The manuscript submitted by Wang et al. presents an experimental study about the fast driving of a spin qubit located in a germanium quantum dot. The QD device is fabricated based on a germanium hut wire with top gates to localize quantum dots along the wire. Wang et al. uses Pauli spin blockade in a multi-hole regime to read-out spin-orbit states manipulation. The authors have studied two electrically driven spin resonance (ESDR) and have shown for one of the resonance an ultra-fast Rabi frequency of 542MHz. They also studied the free evolution and the dephasing of this ultra-fast resonance. Finally, Wang et al. attributes the ultra-fast Rabi frequency to an unusually short spin-orbit length of 2.3nm.

While the experimental results are appealing at a first glance, the theoretical explanation given by the authors about the ultra-fast Rabi oscillation due to an unusually short spin-orbit length seems difficult to apply to the experimental results (see following explanation). Moreover a spin-orbit qubit in a Ge hut wire has already been demonstrated by Watzinger et al. in 2018 and published in Nature Communications. Considering these two last aspects I cannot recommend the publication of the present manuscript in Nature Communications.

Response: We are glad that the reviewer think the experiment result is appealing. We also agree that the explanations we gave in the previous manuscript was not adequate. Therefore we have now developed a new effective two-hole model to describe our experimental observation, and the simulation results provide quite good fit to our experimental results. Below we give a point-by-point response to the comments and suggestions by the reviewer.

We are indeed not the first to study spin qubits in Ge hut wires. However, compared to the work by Watzinger et al. in 2018, where a hole spin qubit in Ge is demonstrated, our work shows much faster Rabi operation with a strong spin-orbit coupling strength arising from the very different confinement thanks to the additional middle gate induced tunable inter-dot coupling. The ultrafast spin control we have demonstrated suggests a strong potential of hut wire Ge hole spin qubits for scalable high-fidelity qubit control. This was

not demonstrated before and should be interesting to the readers. We believe the dramatically faster Rabi frequency and the associated innovations do meet the requirements of novelty and importance by Nature Communications.

Explanation:

1. *From Fig.S2, the EDSR spectrum is complex as it presents a lot of resonances, which indicates that the simple picture of a (1,1)/(0,2) transition cannot be applied here (certainly low level splitting are present in one of the dots). Consequently it is hard to know which states are driven for the EDSR A and B.*

Response: After some explorations, we have developed our model based on effective (1,1)-(2,0) transitions, and the spectrum produced by this model fits the major features of the experimental observations quite well. We have added a detailed discussion of this model in the Supplementary Materials (Sec. 2).

“We measure the EDSR signal by observing the variation in the leakage current in the PSB regime. When EDSR flips a spin, PSB is lifted and the current starts to flow, until another hole with the right spin state blocks the transport again. As such, EDSR leads to an increase in the leakage current. Notice that if none of the hole spins are flipped when driven, PSB cannot be lifted and leakage current would not change. Thus the change in PSB leakage current is a clear indication that a spin-flip transition has happened.

In our measurement, we apply a continuous microwave pulse to one of the gates (R). The electric field E_{ac} that drives EDSR can be estimated from Fig. S8c & d by modeling our device and inputting the a.c. voltage V_{ac} from the microwave source. Under different magnetic fields and microwave frequencies, more than 5 oblique lines with anti-crossings are obtained at a low microwave power $P = -15$ dBm (Fig. S2a) indicating that the microwave is on resonance with a particular spin-flip transition. In the spectroscopy, drifts in the transport current account for current variation along the longitudinal axis while the step-like signal along the horizontal axis is attributed to the attenuation difference of the circuit.

The most commonly observed Pauli Spin Blockade happens near the (0,2) – (1,1) transition, while our DQD contains roughly five to ten holes in each dot. Nevertheless, we find that an effective two-hole model (for valence holes involved in the observed PSB)

produces a spectrum that fits our observation quite well. In this model, we include basis states of as single-dot singlet, S_{02} , and four two-hole states in the (1,1) regime: $|\downarrow\uparrow\rangle$, $|\uparrow\downarrow\rangle$, $|\uparrow\uparrow\rangle$ (T_+) and $|\downarrow\downarrow\rangle$ (T_-). Here, only the triplets in the (1,1) regime are considered since T_{02} has much higher energy than S_{02} due to Pauli exclusion principle. The corresponding two-hole Hamiltonian can be written as

$H_{DQD} =$

$$\begin{pmatrix} -\varepsilon & t/\sqrt{2} & -t/\sqrt{2} & -\Delta_{SO}^* & \Delta_{SO} \\ t/\sqrt{2} & 1/2(g_1 - g_2)\mu_B B & 0 & 0 & 0 \\ -t/\sqrt{2} & 0 & -1/2(g_1 - g_2)\mu_B B & 0 & 0 \\ -\Delta_{SO} & 0 & 0 & -1/2(g_1 + g_2)\mu_B B & 0 \\ \Delta_{SO}^* & 0 & 0 & 0 & 1/2(g_1 + g_2)\mu_B B \end{pmatrix},$$

where g_1 and g_2 are the g-factors of the hole spin in the two dots respectively, t is the spin-independent tunnel coupling between the two dots, Δ_{SO} is the spin-flip tunnel coupling between $|\uparrow\uparrow\rangle$ / $|\downarrow\downarrow\rangle$ and S_{02} , and ε is the detuning of $S_{11} = 1/\sqrt{2}(|\uparrow\downarrow\rangle + |\downarrow\uparrow\rangle)$ with respect to S_{02} . The calculated eigenvalues of the matrix are given in Fig. S2b, showing the five eigenstates as a function of the magnetic field B. The spectral curves arising from the transitions between any two states can be mapped to this energy spectrum. Energetically, spin blockade should happen when the low-energy (1,1) triplet $|\downarrow\downarrow\rangle$ is occupied, and spin-flip transition from this state to any other would lead to an increase in current and thus an EDSR signal. The high-energy triplet $|\uparrow\uparrow\rangle$ is usually unoccupied, especially at higher magnetic field, thus transitions originating from it are generally not observed experimentally. On the other hand, at a very low field it does mix with S_{02} significantly by spin-flip tunneling, such that a seemingly two-spin-flip transition can be seen in the experimental measurement, though that particular signal quickly faded away as the magnetic field is increased.

When operating the spins in experiment, the working position is fixed at a large value of detuning (point M in Fig. 1b) deeply in the Coulomb blockade regime, instead of the low-detuning regime ($\varepsilon \sim 0$) where the EDSR spectrum is measured. To clarify the manipulated spin states more clearly, we perform the spectral calculation in Fig. S2 c & d by assuming $\varepsilon \sim 100 \mu\text{eV}$. The linear resonances can be achieved in this case where rotation of a single spin (either left or right) can be performed for coherent control (Fig. S2 c).”

Revised Figure S2: EDSR spectrum. **a.** Simulated and experimental results of the EDSR spectrum. During spin operation, qubit frequencies of both mode A (yellow) and mode B (purple) are fixed around 8 GHz, which are far from any anti-crossing. In the left panel of **a**, calculated seven transitions between $|T_{\pm}\rangle$ (four colored and three grey curves) and other four states are mapped as a function of magnetic field B . Four colored resonance are indicated by arrows in **b**. Eigenstates, as well as mixing, of these five states vary with the magnetic field. Parameters used for simulation are $g_L = 7$, $g_R = 3.95$, $2t = 14 \mu\text{eV}$, $\varepsilon = 1.5 \mu\text{eV}$ and $|\Delta_{S0}| = 3.8 \mu\text{eV}$. **c & d.** Simulated spectrum and eigenstates at a large detuning $\varepsilon = 100 \mu\text{eV}$. The coherent control of the left (right) spin is performed in the yellow (purple) resonance.

2. Still from Fig.S2 the EDSR line A seems to anti-cross with the EDSR line just above, indicating that one of the states involved in the EDSR A is interacting with another

level of the double dot. Consequently EDSR A is not associated to the driving of a simple spin splitted orbital state in one of the dot.

Response: To highlight the complex physical picture of the EDSR spectroscopy in overhead, we have include the following part in the last paragraphs on page 3 of Supplementary Material.

“...The spectral curves arising from the transitions between any two states can be mapped to this energy spectrum. Energetically, spin blockade should happen when the low-energy (1,1) triplet $|\downarrow\downarrow\rangle$ is occupied, and spin-flip transition from this state to any other would lead to an increase in current and thus an EDSR signal. The high-energy triplet $|\uparrow\uparrow\rangle$ is usually unoccupied, especially at higher magnetic field, thus transitions originating from it are generally not observed experimentally. On the other hand, at a very low field it does mix with S_{02} significantly by spin-flip tunneling, such that a seemingly two-spin-flip transition can be seen in the experimental measurement, though that particular signal quickly faded away as the magnetic field is increased.”

3. The authors indicates a g -factor of 4.5 and 3 for the EDSR A and B, respectively. To my knowledge, these g -factor are far from the heavy-hole g -factor expected in Germanium.

Response: As mention in the response to Reviewer 1 and Reviewer 2, our new two-hole model predicts g -factors of 7 and 3.95 instead of 4.5 and 3, although they are still far from the expected g -factor in Ge. Thus we have added the discussion in the 2nd paragraph on page 5 of Supplementary Material. “Here, we obtain the two g -factors as $g_L = 7$ and $g_R = 3.95$. For heavy holes in Ge hut wire, one expects a small in-plane g -factor of 0.2 and a large out-of-plane g -factor of 21.4^{S1}. In our model, we assume two different g -factors for the spins in the left dot and the right dot respectively. We believe this difference can be attributed to the unequal hole occupations between the two dots. Our observations seem to indicate that fewer holes occupy the left dot compared to the right dot (Fig. S1b & Fig. S8b). It is thus quite probable that the wave function of the manipulated spin differs in the two dots. Moreover, a recent preprint^{S2} shows a large out-of-plane g -factor of 15.7 and very different g -factors due to a different hole filling. In short, the g -factor difference between mode A and mode B is quite understandable considering their different occupations and states involved.” Besides, the observed g -factor of Ge quantum dots in

Ge/Si coreshell nanowire are 0.2~4 while an in-plane g -factor of 0.2-0.3 and an out-plane g -factor of 7.5 are measured in planer quantum dots. In HW system, hole g -factor up to 4.4 were found. Among these hole quantum system, planar quantum dots and HW quantum dots have been identified as HH system with tiny LH admixture (*Nature Reviews Materials* 2020, 1-18).

4. *From all the above remarks, I do not understand how the authors can claim (even qualitatively) that they are driving an HH spin transition. From the complexity of the EDSR spectrum, it appears clear that the level splitting in one of the QD (at least) is small. From the small g -factor, it is also clear that the states involved in the EDSR resonances have mixed heavy and light hole character. The Rabi frequency observed is then certainly coming from a complex interplay between spin-orbit interaction and multi-level structure with heavy and light admixture of the QD under consideration. Consequently, I cannot agree with the authors on the description of the EDSR mechanism has just the effect of a Rashba-type SOI acting on spin 3/2 heavy hole.*

Response: The nature of holes in our system is described in Section 9 of Supplementary Material, in which the calculation results show that our manipulated spins correspond to nearly pure heavy hole states, which is consistent with the results of Ref [29] and Ref [33]. As mentioned above, our model describes the EDSR spectrum well and the fast Rabi frequency of each spin comes from the small spin-orbit length arising from our Ge/Si material. Therefore, we believe the claim for HH spin transition in our system is reasonable.

29. Watzinger, H. et al. Heavy-hole states in germanium hut wires. *Nano lett.* **16**, 6879-6885 (2016).

33. Katsaros G. et al. Zero Field Splitting of Heavy-Hole States in Quantum Dots. *Nano Lett.* **20**, 5201-5206 (2020).

REVIEWER COMMENTS

Reviewer #1 (Remarks to the Author):

In their rebuttal letter, the authors addressed the remarks I had, most of which were shared by the other two reviewers, and implemented important changes in the manuscript and supplementary information. In particular, I want to emphasize the important theoretical effort to understand the physical nature of their two modes A and B, which was in my opinion one of the weak points of their earlier manuscript. With this added content, I can now recommend publication in Nature Communication, pending a few more comments and questions on their modifications:

- While the two-spin model fits the experimental spectroscopy quite well, there is one line absent in the simulation (see Fig 1 below). Do the authors have a hint about the nature of this transition? Is it a two-photon mechanism, or does it involve a higher orbital state?
- If I understand the revised Fig. S2 correctly, the coherent control of the spin is performed in a totally different regime than the one presented in the spectroscopy in the main paper. Is there a reason why the spectroscopy could not be performed at point M?
- The authors state that “The g-factor for a hole spin depends on control parameters such as the inter-dot tunnel coupling”. I do not agree with this claim, but maybe I misunderstand what it means. While the g-factor of a single-spin may depend on the shape of the quantum dot and its occupation, it is independent of the inter-dot tunnel coupling. However, the energy splitting of the system (and its dependence on B_z) may depend on each single-spin g-factors g_L and g_R and on the inter-dot tunnel coupling.
- I think the added discussion on the modulation of the g-factors with the electrostatic environment is quite interesting. Can the authors give an explanation on the faster oscillations obtained at M1 with respect to M2?
- I agree with the second referee about the relevance of the Fig. S10 plot. In my opinion, T_2^* is not an appropriate metric to benchmark the different works among the community. Rather, the coherence time during a manipulation (T_{2Rabi}) should be used. For example, in the case of a noisy drive, one could have a good T_2^* and f_{Rabi} , but the qubit would not have a good Q-factor. That being said, I admit that T_2^* values are much easier to find in the community, often without any

indication of the power used to obtain them. Maybe a short discussion on the different metrics used to benchmark the qubits and their flaws could be added to the Supplementary section 8.

Reviewer #2 (Remarks to the Author):

I am pleased about the revised manuscript, supp info and the rebuttal, and heartily recommend it for publication as is.

Floris Zwanenburg

Reviewer #3 (Remarks to the Author):

Review II on “Ultrafast Operation of a Hole spin qubit in a Germanium Quantum dot” by Wang et al.

Wang et al. have added some supplementary material and change the main text accordingly. The main message of the manuscript didn't change and to me the data and theory provided did not change my first impression on the manuscript.

Here below I listed

“with a strong spin-orbit coupling strength arising from the very different confinement thanks to the additional middle gate induced tunable inter-dot coupling.”

- There is no experimental data indicating that the confinement is largely changed compared to Watzinger et al. thanks to the middle gate.

“Moreover, a faster Rabi oscillation can be achieved by changing the inter-dot coupling”

- Could the authors argue about this point? Because the Rabi frequency presented here are measured deep in Coulomb Blockade for which in principle single spin transition are considered (as claimed by the authors themselves), then the inter-dot coupling does not play any role here. The statement is to me wrong.

The ultrafast spin control we have demonstrated suggests a strong potential of hut wire Ge hole spin qubits for scalable high-fidelity qubit control.

- The verb “suggest” indicates effectively that nothing in this manuscript is demonstrating high fidelity...

We believe the dramatically faster Rabi frequency and the associated innovations do meet the requirements of novelty and importance by Nature Communications.

- Which “associated innovations”

The authors present a simulated EDSR spectrum in order to explain their data. For me there is two points I would like to raise here:

- 1- In the figure below (bottom left) I superimposed colored lines on the experimental data and then did a mirror along the vertical axis in order to compare the experimental EDSR lines I see and the calculated spectrum. I can agree with the authors that the spectrum looks close but still there is some discrepancy. 2 EDSR lines totally not captured by the simulation (black dashed lines) and anticrossing not captured also (between red and yellow

lines close to zero field and around 6 GHz between the 3 EDSR lines visible in the experiment, only two in the experiment).

- 2- Since the simulated spectrum is not in total agreement with the experimental data, the authors could have done complementary measurements to convince future readers. For example, the EDSR spectrum could have been realized even with a pulse sequence between coulomb blockade and spin blockade. Like that, we would have obtain the excitation spectrum at the qubit manipulation point (which would be more convincing that just a simulation as FigS2 c)). Similarly an EDSR spectrum at fix magnetic field depending on the detuning could have also helped.

3. The authors indicate a g -factor of 4.5 and 3 for the EDSR A and B, respectively. To my knowledge, these g -factors are far from the heavy-hole g -factor expected in Germanium.

Response: As mentioned in the response to Reviewer 1 and Reviewer 2, our new two-hole model predicts g -factors of 7 and 3.95 instead of 4.5 and 3, although they are still far from the expected g -factor in Ge. Thus we have added the discussion in the 2nd paragraph on page 5 of Supplementary Material. "Here, we obtain the two g -factors as $g_L = 7$ and $g_R = 3.95$. For heavy holes in Ge hut wire, one expects a small in-plane g -factor of 0.2 and a large out-of-plane g -factor of 21.4_{S1}. In our model, we assume two different g -factors for the spins in the left dot and the right dot respectively. We believe this difference can be attributed to the unequal hole occupations between the two dots. Our observations seem to indicate that fewer holes occupy the left dot compared to the right dot (Fig. S1b & Fig. S8b). It is thus quite probable that the wave function of the manipulated spin differs in the two dots. Moreover, a recent preprint_{S2} shows a large out-of-plane g -factor of 15.7 and very different g -factors due to a different hole filling. In short, the g -factor difference between mode A and mode B is quite understandable considering their different occupations and states involved." Besides, the observed g -factor of Ge quantum dots in 41/41 Ge/Si coreshell nanowire are 0.2~4 while an in-plane g -factor of 0.2-0.3 and an out-of-plane g -factor of 7.5 are measured in planar quantum dots. In HW system, hole g -factor up to 4.4 were found. Among these hole quantum system, planar quantum dots and HW quantum dots have been identified as HH system with tiny LH admixture (*Nature Reviews Materials* 2020, 1-18).

And

4. From all the above remarks, I do not understand how the authors can claim (even qualitatively) that they are driving an HH spin transition. From the complexity of the EDSR spectrum, it appears clear that the level splitting in one of the QD (at least) is small. From the small g -factor, it is also clear that the states involved in the EDSR resonances have mixed heavy and light hole character. The Rabi frequency observed is then certainly coming from a complex interplay between spin-orbit interaction and multi-level structure with heavy and light admixture of the QD under consideration. Consequently, I cannot agree with the authors on the description of the EDSR mechanism as just the effect of a Rashba-type SOI acting on spin 3/2 heavy hole.

Response: The nature of holes in our system is described in Section 9 of Supplementary Material, in which the calculation results show that our manipulated spins correspond to nearly pure heavy hole states, which is consistent with the results of Ref [29] and Ref [33]. As mentioned above, our model describes the EDSR spectrum well and the fast Rabi frequency of each spin comes from the small spin-orbit length arising from our Ge/Si material. Therefore, we believe the claim for HH spin transition in our system is reasonable. 29. Watzinger, H. et al. Heavy-hole states in germanium hut wires. *Nano Lett.* **16**, 6879- 6885 (2016). 33. Katsaros G. et al. Zero Field Splitting of Heavy-Hole States in Quantum Dots. *Nano Lett.* **20**, 5201-5206 (2020).

→ I do understand that theory presented in S9 and some experimental results demonstrate that the hole is mainly of heavy-hole type in hut wire. However it is

known that disorder, strain, hole filling.... Can change drastically the hole nature in a real device. Then here I would like to raise two points:

- To be convincing with the supplementary theory S9, the authors could have calculated the g-factors expected and show that they match the experimental data. But I guess this is not the case.
- To be convincing experimentally I would like to see g-factor magnetic field anisotropy, gate dependence for a known number of holes.

About the supplementary S10

Once again, here, the only “qualitative understanding” is not sufficient for the authors to claim that the fast Rabi oscillations are only coming from an ultra-short spin-orbit length. It is known in the community that the electrically driven spin resonance of hole is a complex interplay between two mechanisms namely EDSR and g-tensor modulation (which with the g-factor reported cannot be neglected in the analysis). These two mechanism being hard to capture theoretically in the multi-hole regime with a strong LH/HH mixing, strong electrical field, possibly strain and disorder. Consequently, Figure 1-c cannot be used to illustrate the experimental results here as the contribution from g-tensor contribution to EDSR has not been characterized and cannot be rule-out.

About figure S8

In Fig S8d, I don't understand why the simulation is not giving a gradient of field $E_y(AC)$ along the y axis. I would expect that applying the microwave signal on the right gate will make the left hole moving along the wire. Moreover it would be nice to see the $E_z(ac)$, which will participate to g-tensor modulation.

“Due to the smaller a.c. field in the right dot (see Supplementary Material, Sec. 6), mode B shows a slower speed of Rabi rotation compared to mode A even though the spin-orbit lengths are close”

- This sentence is not complete. The left dot is also not seeing the same direction of microwave field excitation. Then the EDSR mechanism could be different.

Moreover, a faster Rabi oscillation can be achieved by changing the inter-dot coupling or the manipulation position. Such optimization can be achieved by tuning the voltage of gate R. Our test results show that the Rabi frequency can be increased from 63 MHz to 111 MHz by switching the manipulation position from M2 to M1 (Fig. S4 in Supplementary Material). We are thus optimistic that a faster Rabi operation is achievable after optimization.

- Between M1 and M2, the g-factor is changing, then the microwave frequency to apply to drive Rabi oscillation is different. How does the authors know that the microwave power delivered to the device is the same between the point M1 and M2. This could explain the change in Rabi frequency.

To conclude, I would summarize my position as follow: Wang et al. achieve ultra-fast Rabi oscillations in a double quantum dot formed in a germanium hut wire with no clear experimental or theoretical demonstration on their origin. The resonance mechanism is only approached “qualitatively” and deep characterization of the dots g-factor and EDSR mechanism is missing.

Once again, if this manuscript would have been the first one to report spin qubits in hut wire I would support its publication (together with major corrections) but hut wire spin-qubit have already been demonstrated. Thus the lack of convincing demonstration (experimental and theoretical) on the origin of the fast Rabi oscillation reported here makes that I cannot recommend this manuscript for publication in Nature Communications.

Response to Reviewers' Comments

We thank the Reviewers for their time and patience with our manuscript. Below we give a point by point answer to the questions raised by the Reviewers. The comments from the Reviewers are reproduced in *purple color and Calibri font* along with our responses in black text while modified contents are indicated by *blue color and Times New Roman font*.

Response to Reviewer 1

Comments: In their rebuttal letter, the authors addressed the remarks I had, most of which were shared by the other two reviewers, and implemented important changes in the manuscript and supplementary information. In particular, I want to emphasize the important theoretical effort to understand the physical nature of their two modes A and B, which was in my opinion one of the weak points of their earlier manuscript. With this added content, I can now recommend publication in Nature Communication, pending a few more comments and questions on their modifications:

Response: We thank the Reviewer for these positive comments and for acknowledging our theoretical effort in explaining the EDSR spectrum. We address the comments and questions raised by the Reviewer below.

1. While the two-spin model fits the experimental spectroscopy quite well, there is one line absent in the simulation (see Fig 1 below). Do the authors have a hint about the nature of this transition? Is it a two-photon mechanism, or does it involve a higher orbital state?

Figure 1: Copy from Reviewer #1

Response: We thank the Reviewer for pointing out the transition line that was not included in our original model. Indeed, we also noticed this line and another resonance as mentioned by Reviewer #3. Neither was included in our initial model for the sake of simplicity. We have now added an excited orbital state in our model, and the two dashed lines can be mapped out by involving the excited orbital state with two opposite spins, $|\downarrow\uparrow\rangle_e$ and $|\uparrow\downarrow\rangle_e$ in Fig. S2 a & b.

We have modified the description on page 3 & 4 in the revised Supplementary Material for clarity.

“The most commonly observed Pauli Spin Blockade happens near the $(0,2) - (1,1)$ transition, though it has also been observed in multi-electron double dots^{S1-S3} while our DQD contains roughly five to ten holes in each dot, we find that an effective two-hole model (essentially assuming that core holes do not participate in the low energy dynamics related to PSB) produces a spectrum that fits our observation quite well. In this model, we include basis states of single-dot singlet, S_{02} , and six two-hole states in the (11) regime: $|\downarrow\uparrow\rangle$, $|\uparrow\downarrow\rangle$, $|\uparrow\uparrow\rangle$ (T_+) and $|\downarrow\downarrow\rangle$ (T_-) in the ground orbital states, along with two zero-spin states in an excited orbital state $|\uparrow\downarrow\rangle_e$ and $|\downarrow\uparrow\rangle_e$. Here, only the ground triplets in the (11) regime are considered since T_{02} has much higher energy than S_{02} due to Pauli exclusion principle. The excited triplets are not discussed in our model. We also do not consider $|\downarrow\downarrow\rangle_e$ as the transition does not lift PSB and cannot be observed in our experiment. The corresponding two-hole Hamiltonian can be written as

$$H_{DQD} = \begin{pmatrix} \Delta_e - (g_1 - g_2)\mu_B B/2 & 0 & -t_e/\sqrt{2} & 0 & 0 & \Delta_{S0}^{e1*} & -\Delta_{S0}^{e1} \\ 0 & \Delta_e + (g_1 - g_2)\mu_B B/2 & t_e/\sqrt{2} & 0 & 0 & \Delta_{S0}^{e2*} & -\Delta_{S0}^{e2} \\ -t_e/\sqrt{2} & t_e/\sqrt{2} & -\varepsilon & -t/\sqrt{2} & t/\sqrt{2} & \Delta_{S0}^e & -\Delta_{S0} \\ 0 & 0 & -t/\sqrt{2} & -(g_1 - g_2)\mu_B B/2 & 0 & 0 & 0 \\ 0 & 0 & t/\sqrt{2} & 0 & (g_1 - g_2)\mu_B B/2 & 0 & 0 \\ \Delta_{S0}^{e1} & \Delta_{S0}^{e2} & \Delta_{S0} & 0 & 0 & (g_1 + g_2)\mu_B B/2 & 0 \\ -\Delta_{S0}^{e1*} & -\Delta_{S0}^{e2*} & -\Delta_{S0} & 0 & 0 & 0 & -(g_1 + g_2)\mu_B B/2 \end{pmatrix}$$

where g_1 and g_2 are the g-factors of the hole spin in the two dots respectively, t and t_e are the spin-independent tunnel coupling between the two dots, Δ_{S0} is the spin-flip tunnel coupling between $|\uparrow\uparrow\rangle$ / $|\downarrow\downarrow\rangle$ and S_{02} , Δ_{S0}^e is the spin-flip tunnel coupling between excited state and $|\uparrow\uparrow\rangle$ / $|\downarrow\downarrow\rangle$, Δ_e is the energy gap between ground states and excited states, and ε is the detuning of $S_{11} = 1/\sqrt{2}(|\uparrow\downarrow\rangle - |\downarrow\uparrow\rangle)$ with respect to S_{02} .

The calculated eigenvalues of the matrix at point R (i.e. in the PSB regime, with small positive detuning ϵ) are given in Fig. S2b, showing the seven eigenstates as a function of the magnetic field B . The spectral curves arising from the transitions between any two states can be mapped to this energy spectrum. Energetically, spin blockade happens when the low-energy (11) triplet $|\downarrow\downarrow\rangle$ is occupied, and spin-flip transition from this state to any other would lead to a lift of PSB and an increase in current, and thus an EDSR signal. The high-energy triplet $|\uparrow\uparrow\rangle$ is usually unoccupied, especially at higher magnetic field, thus transitions originating from it are generally not observed experimentally. On the other hand, at a very low field it does mix with S_{02} significantly by spin-flip tunneling, such that a second-order two-spin-flip transition can be seen in the experimental measurement, though that particular signal quickly fades away as the magnetic field is increased. Moreover, two excited-state related resonances are included in the spectrum as well (blue and red curves in Fig. S2a).

When we measure Rabi oscillations and Ramsey fringes of the spins, the working position is fixed at a large value of detuning (point M in Fig. 1b) deep in the Coulomb blockade regime, instead of point R in the low-detuning regime ($\epsilon \sim 0$) where the EDSR spectrum Fig.1d is measured. To clarify the manipulated spin states more clearly, we also perform the spectral calculation in Fig. S2 c & d by assuming $\epsilon \sim -100 \mu\text{eV}$, where the (1,1) states are well decoupled from the S_{02} singlet.

Figure S2: EDSR spectrum. **a.** Simulated and experimental results of the EDSR spectrum. Left panel: calculated resonances between $|\downarrow\downarrow\rangle$ and other six states (two dashed and four solid curves). These colored resonance are indicated by arrows in **b**. Eigenstates, as well as mixing, of these seven states vary as a function of magnetic field. Parameters used for simulation are $g_L = 7$, $g_R = 3.95$, $2t = 8 \mu\text{eV}$, $2t_e = 8 \mu\text{eV}$, $\varepsilon = 5 \mu\text{eV}$, $|\Delta_{S0}| = 1 \mu\text{eV}$, $\Delta_e = 12 \mu\text{eV}$, $|\Delta_{S0}^{e1}| = 3 \mu\text{eV}$ and $|\Delta_{S0}^{e2}| = 0.5 \mu\text{eV}$. **c & d.** Simulated spectrum and eigenstates at a large detuning $\varepsilon = -100 \mu\text{eV}$ (Coulomb blockade regime). The coherent control of the left (right) spin is performed at the yellow (purple) resonance.

”

The related Figure 1e & 1f in the main text are modified as well.

“

Figure 3 and New Figure 1: e, Calculated EDSR spectrum from our effective two-hole model with g -factors of left and right dot of $g_L = 7$ and $g_R = 3.95$. The **two highlighted resonances** (purple, yellow) correspond to spin-flips between $|\downarrow\downarrow\rangle$ and $|\downarrow\uparrow\rangle/|\uparrow\downarrow\rangle$, indicated by the arrows in the energy level diagram **f**. **Other related resonances** are colored in grey in **e** (see details in Supplementary Material, Sec.2, EDSR spectrum).”

2. If I understand the revised Fig. S2 correctly, the coherent control of the spin is performed in a totally different regime than the one presented in the spectroscopy in the main paper. Is there a reason why the spectroscopy could not be performed at point M?

Response: Indeed, the manipulation point is different from the point at which the EDSR spectrum is obtained. This is an important question. The limitation associated with this procedure is that readout needs to be performed in the regime of PSB where we can distinguish states by electric current as long as the spin is flipped. If the spectrum is measured at the manipulation point outside the triangle, no current change can be detected due to Coulomb blockade. If the question is why the spectroscopy could not be performed with a sequence and measured at point R, the reason is that SNR (Signal To Noise Ratio) of spin-flip after coherent manipulation would be dramatically reduced and a clear spectroscopy is hard to obtain.

3. The authors state that “The g -factor for a hole spin depends on control parameters such as the inter-dot tunnel coupling”. I do not agree with this claim, but maybe I misunderstand what it means. While the g -factor of a single-spin may depend on the

shape of the quantum dot and its occupation, it is independent of the inter-dot tunnel coupling. However, the energy splitting of the system (and its dependence on B_z) may depend on each single-spin g -factors g_L and g_R and on the inter-dot tunnel coupling.

Response: We thank the Reviewer for pointing out the clumsy phrasing in the previous draft. What we tried to convey is that the slope of the resonance is a function of the voltage on gate M. We have revised this sentence in the last paragraph on page 6 in Supplementary Material for clarity.

“The condition of resonances depends on control parameters such as the inter-dot tunnel coupling or the electrostatic field, which can be seen in Fig. S3, where the oblique resonance varies as a function of the middle gate voltage.”

4. I think the added discussion on the modulation of the g -factors with the electrostatic environment is quite interesting. Can the authors give an explanation on the faster oscillations obtained at M1 with respect to M2?

Response: We are grateful to the Reviewer for their interest. In Supplementary Material, Figure S4 demonstrates that the Rabi rotation speed of our qubit can be further improved and related work is ongoing. The mechanism behind this phenomenon is still unclear, though we plan to explore it in a forthcoming work (in preparation). At the moment we are not sure if it can be attributed to a stronger spin-orbit coupling at smaller detuning.

5. I agree with the second referee about the relevance of the Fig. S10 plot. In my opinion, T_2^ is not an appropriate metric to benchmark the different works among the community. Rather, the coherence time during a manipulation ($T_{2\text{Rabi}}$) should be used. For example, in the case of a noisy drive, one could have a good T_2^* and f_{Rabi} , but the qubit would not have a good Q -factor. That being said, I admit that T_2^* values are much easier to find in the community, often without any indication of the power used to obtain them. Maybe a short discussion on the different metrics used to benchmark the qubits and their flaws could be added to the Supplementary section 8.*

Response: We are grateful for this useful suggestion. We have included a discussion relevant to the use of T_2^* on page 19 in the Supplementary Material.

“There are many ways to evaluate the quality of a qubit or its manipulation. Two good ways are gate tomography and fidelity via randomized benchmarking. Tomography is a good gauge of the system and its coherence terms, and has been used in some previous works, though it takes a great deal of time to obtain the density matrix elements. Gate fidelity is comparatively easier to obtain and is measured in many experiments. Here we would like to evaluate how many operations can be performed before a qubit decoheres. Thus we focus on the x-axis rotation speed and coherence time of different spin qubits in previous literatures. Please note that we use the values of T_2^* only for qualitative comparison, because it may also depend on the microwave power which can induce noise during a measurement. In short, our simple comparison here is only meant to show the general characteristics of different spin qubits.”

Response to Reviewer 2

Comments: I am pleased about the revised manuscript, supp info and the rebuttal, and heartily recommend it for publication as is.

Response: We want to thank the reviewer for their continued efforts and we are grateful for this enthusiastic recommendation.

Response to Reviewer 3

Comments: Wang et al. have added some supplementary material and change the main text accordingly. The main message of the manuscript didn't change and to me the data and theory provided did not change my first impression on the manuscript. Here below I listed:

“with a strong spin-orbit coupling strength arising from the very different confinement thanks to the additional middle gate induced tunable inter-dot coupling.”

There is no experimental data indicating that the confinement is largely changed compared to Watzinger et al. thanks to the middle gate.

Response: As shown in Fig. 4 a & b there are only two gates with negative voltages in Watzinger et al.'s work, while positive voltages are applied to gate L and gate R in our case due to the existence of the middle gate (Fig. 4 c & d). A singlet-triplet splitting up to ~ 1 meV and the Rabi frequency approaching 140 MHz are mentioned in their device with gate electrodes exceeding 60 nm and g-factor ~ 2 . Comparatively, both our hole filling and confinement are quite different from theirs considering the large difference of the applied voltage and smaller gate width of 30 nm, and the g-factor of our wire is larger than 3.9. Consequently, we obtain an energy splitting of 3 meV (noted as $\hbar\omega_y$ in the caption of Figure 4 in the main text) and a Rabi frequency of 540 MHz. Furthermore, the electric field in the device is also not mentioned in their paper, making it even more difficult to have a direct comparison. Nevertheless, according to our equation (2) $hf_{\text{Rabi}} = g\mu_B B \cdot \frac{l_{\text{dot}}}{l_{\text{so}}} \cdot \frac{eE_{\text{ac}}l_{\text{dot}}}{\hbar\omega_y}$, we speculate that our faster manipulation can be attributed to a much smaller spin-orbit length for our device if their electric field strengths are close to ours.

Figure 4: a & b Device and studied current triangles in Watzinger's work compared to our results in c & d.

“Moreover, a faster Rabi oscillation can be achieved by changing the inter-dot coupling”

Could the authors argue about this point? Because the Rabi frequency presented here are measured deep in Coulomb Blockade for which in principle single spin transition are considered (as claimed by the authors themselves), then the interdot coupling does not play any role here. The statement is to me wrong.

Response: We thank the Reviewer for spotting this oversight. We have checked the point and agree that the Rabi frequency has a weak dependence on the inter-dot coupling ($2t$) when the spin is coherently manipulated at large detuning instead of small ones. At large detuning of $\varepsilon = -100 \mu\text{eV}$ (Fig. 5b) the eigenstates vary slightly as a function of $2t$ while they would have a strong dependence on it at a small detuning of $\varepsilon = 5 \mu\text{eV}$ (Fig. 5a) which led to our erroneous statement. We have revised the sentence as below on page 10 in the revised main text:

“Moreover, a faster Rabi oscillation can be achieved by changing the inter-dot coupling or the manipulation position. Such optimization can be achieved by tuning the voltage of gate R .”

Figure 5: Energy level at $\varepsilon = 5 \mu\text{eV}$ (a) and at $\varepsilon = -100 \mu\text{eV}$ (b). Other parameters: $B = 0.1 \text{ T}$, $g_L = 7$, $g_R = 3.95$, $2t = 8 \mu\text{eV}$, $2t_e = 8 \mu\text{eV}$, $\varepsilon = 5 \mu\text{eV}$, $|\Delta_{S0}| = 1 \mu\text{eV}$, $\Delta_e = 12 \mu\text{eV}$, $|\Delta_{S0}^{e1}| = 3 \mu\text{eV}$ and $|\Delta_{S0}^{e2}| = 0.5 \mu\text{eV}$.

“The ultrafast spin control we have demonstrated suggests a strong potential of hut wire Ge hole spin qubits for scalable high-fidelity qubit control.”

The verb “suggest” indicates effectively that nothing in this manuscript is demonstrating high fidelity...

Response: Our result of a quality factor of ~ 45 is the highest among nanowire spin qubits, though it is indeed smaller compared to some quantum dot systems based on two-

dimensional gases as shown in Fig. S10. Besides, Philippopoulos et al. [*PRB* 101, 115302 (2020)] reported that with the p-atomic-orbital wave function of hole states in Ge, the contact hyperfine interaction vanishes completely and the anisotropic hyperfine interaction (dipole-dipole and angular-momentum terms) dominates, leading to a long coherence time. Ge materials can also be further purified into a nuclear-spin free host to improve the coherence times of spin qubits in the future. Considering the potential for even faster control speed and possible long coherence time, we would argue that high-quality qubit control in Ge hut nanowires are as promising a path forward as other types of nanowires and materials and deserves further studies. From a broader perspective, with no spin qubits having been established as the unquestioned candidate for future scalable quantum computers (spin qubits in Si conduction band is the unquestioned leader in the race but even that system faces the challenge of valley-orbit physics, which is absent in Ge hole systems), we believe the exploration of alternative qubit systems is scientifically important.

Reference:

Philippopoulos, P., Chesi, S. and Coish W. A., First-principles hyperfine tensors for electrons and holes in GaAs and silicon. *Phys. Rev. B* **101**, 115302 (2020)

“We believe the dramatically faster Rabi frequency and the associated innovations do meet the requirements of novelty and importance by Nature Communications.”

Which “associated innovations”

Response: This is an important question. Strong spin-orbit coupling, with such a small spin-orbit length of around 1.5 nm, has not been reported before, and we believe this accounts for our fast control. In addition, our rotation speed (Rabi frequency) is five times larger compared to Watzinger et al.’s work, which is technically difficult once the Rabi frequency exceeds 100 MHz. As mentioned above, we fabricated shallower electrodes to confine quantum dot and obtained an energy splitting of 3 meV, which help to reach the 540 MHz Rabi frequency. Our spin-orbit length of 1.5 nm in quantum dot is the smallest one up to date.

The authors present a simulated EDSR spectrum in order to explain their data. For me there is two points I would like to raise here:

Figure 6: Copy from Reviewer #3

1. In the figure below (bottom left) I superimposed colored lines on the experimental data and then did a mirror along the vertical axis in order to compare the experimental EDSR lines I see and the calculated spectrum. I can agree with the authors that the spectrum looks close but still there is some discrepancy. 2 EDSR lines totally not captured by the simulation (black dashed lines) and anticrossing not captured also (between red and yellow lines close to zero field and around 6 GHz between the 3 EDSR lines visible in the experiment, only two in the experiment).

Response: We thank the Referee for this careful examination. As it has been mentioned in the first question of Reviewer #1, we have made relevant revisions to increase the clarity of our manuscript. Specifically, to account for more transitions implies that we need to include more energy levels in our model. Indeed, by adding one excited orbital level [which is energetically quite close to the ground orbital level. This is not really surprising considering that our dot occupation is $(m+1, n+1)$ instead of $(1,1)$], and modifying the parameters slightly, we can account for both those missing dashed lines. In the revised manuscript and Supplementary Material we have updated our model accordingly.

Figure 7 and New Figure S2: EDSR spectrum. **a.** Simulated and experimental results of the EDSR spectrum. **Left panel:** calculated resonances between $|\downarrow\downarrow\rangle$ and other six states (two dashed and four solid curves). These colored resonance are indicated by arrows in **b.** Eigenstates, as well as mixing, of these seven states vary as a function of magnetic field. Parameters used for simulation are $g_L = 7$, $g_R = 3.95$, $2t = 8 \mu\text{eV}$, $2t_e = 8 \mu\text{eV}$, $\varepsilon = 5 \mu\text{eV}$, $|\Delta_{S0}| = 1 \mu\text{eV}$, $\Delta_e = 12 \mu\text{eV}$, $|\Delta_{S0}^e| = 3 \mu\text{eV}$ and $|\Delta_{S0}^{e2}| = 0.5 \mu\text{eV}$.

2. Since the simulated spectrum is not in total agreement with the experimental data, the authors could have done complementary measurements to convince future readers. For example, the EDSR spectrum could have been realized even with a pulse sequence between coulomb blockade and spin blockade. Like that, we would have obtain the excitation spectrum at the qubit manipulation point (which would be more convincing that just a simulation as FigS2 c)). Similarly an EDSR spectrum at fix magnetic field depending on the detuning could have also helped.

Response: We appreciate this suggestion, though we believe our simulation is in good agreement with our data. As mentioned in the response to the second question of Reviewer #1, we cannot directly measure the spectrum with a pulse sequence due to the low SNR. However, we have in fact measured Rabi oscillations as a function of microwave frequency with a pulse sequence and extracted the qubit frequency as shown in Fig. S4b. We find that the qubit frequency shifts as a function of detuning (pulse height) which corresponds to the variation of EDSR spectrum at fix magnetic field. Here the qubit frequency shows where spin resonance occurs.

Figure 8 and Figure S4b: Qubit frequency shifting as a function of pulse height.

3. “The authors indicates a g -factor of 4.5 and 3 for the EDSR A and B, respectively. To my knowledge, these g -factor are far from the heavy-hole g -factor expected in Germanium.

Response: As mention in the response to Reviewer 1 and Reviewer 2, our new two-hole model predicts g -factors of 7 and 3.95 instead of 4.5 and 3, although they are still far from the theoretical g -factor in Ge. For this reason we have added the discussion in the 2nd paragraph on page 5 of Supplementary Material. “Here, we obtain the two g -factors as $g_L = 7$ and $g_R = 3.95$. For heavy holes in a Ge hut wire, one expects a small in-plane g -factor of 0.2 and a large out-of-plane g -factor of 21.451. It must be noted, however that (i) the latter is a theoretical value that has never been observed experimentally (ii) many-body effects on the hole g -factor have not been considered theoretically to date. In our model, we assume two different g -factors for the spins in the left dot and the right dot respectively. We believe this difference can be attributed to the unequal hole occupations between the two dots. Our observations seem to indicate that fewer holes occupy the left dot compared to the right dot (Fig. S1b & Fig. S8b). It is thus quite probable that the wave function of

the manipulated spin differs in the two dots. Moreover, a recent preprint^{S2} shows a large out-of-plane g -factor of 15.7 and very different g -factors due to a different hole filling. In short, the g -factor difference between mode A and mode B is quite understandable considering their different occupations and states involved.” Besides, the observed g -factor of Ge quantum dots in 41 / 41 Ge/Si coreshell nanowire are $0.2 \sim 4$ while an in-plane g -factor of 0.2-0.3 and an outplane g -factor of 7.5 are measured in planer quantum dots. In HW system, hole g -factors up to 4.4 were found. Among these hole quantum system, planar quantum dots and HW quantum dots have been identified as HH system with tiny LH admixture (Nature Reviews Materials 2020, 1-18).”

And 4. “From all the above remarks, I do not understand how the authors can claim (even qualitatively) that they are driving an HH spin transition. From the complexity of the EDSR spectrum, it appears clear that the level splitting in one of the QD (at least) is small. From the small g -factor, it is also clear that the states involved in the EDSR resonances have mixed heavy and light hole character. The Rabi frequency observed is then certainly coming from a complex interplay between spin-orbit interaction and multi-level structure with heavy and light admixture of the QD under consideration. Consequently, I cannot agree with the authors on the description of the EDSR mechanism has just the effect of a Rashba-type SOI acting on spin 3/2 heavy hole.

Response: The nature of holes in our system is described in Section 9 of Supplementary Material, in which the calculation results show that our manipulated spins correspond to nearly pure heavy hole states, which is consistent with the results of Ref [29] and Ref [33]. As mentioned above, our model describes the EDSR spectrum well and the fast Rabi frequency of each spin comes from the small spin-orbit length arising from our Ge/Si material. Therefore, we believe our understanding of the spin transition in our system as being HH is the most reasonable interpretation. 29. Watzinger, H. et al. Heavy-hole states in germanium hut wires. Nano lett. 16, 6879-6885 (2016). 33. Katsaros G. et al. Zero Field Splitting of Heavy-Hole States in Quantum Dots. Nano Lett. 20, 5201-5206 (2020).”

I do understand that theory presented in S9 and some experimental results demonstrate that the hole is mainly of heavy-hole type in hut wire. However it is known that disorder, strain, hole filling... Can change drastically the hole nature in a real device. Then here I would like to raise two points:

- To be convincing with the supplementary theory S9, the authors could have calculated the g-factors expected and show that they match the experimental data. But I guess this is not the case

Response: We agree that calculated g-factor cannot match the experiment perfectly. A result of $g_{\perp} \approx 6\kappa + \frac{27q}{2} + g_c \approx 15$ can be obtained based on our calculation, which has been shown in Ref. 29 (*Nano lett.* 16, 6879- 6885 (2016)). It is reported that the remaining deviation is mainly due to the following reason. First, given the small height of the HW and relative small band offset between Ge and Si, the hole wave function should leak into the surrounding Si, which in turn leads to a reduction of g_{\perp} . Second, the parameters of bulk Ge are used for simplicity and the confinement and strain can lead to a substantial rescaling of the effective band structure parameters. Finally, while we believe the assumption of an infinite HW with a rectangular cross-section is a reasonable approximation, variations to the confinement potential, especially in its shape, can lead to additional corrections. Taking all these elements into account is beyond the scope of the work and requires extensive numerics, especially considering that our dot contains multiple holes and many-body effects are expected to be strong in the hole systems.

Reference:

Watzinger, H. et al. Heavy-hole states in germanium hut wires. *Nano lett.* **16**, 6879-6885 (2016).

- To be convincing experimentally I would like to see g-factor magnetic field anisotropy, gate dependence for a known number of holes.

Response: We agree with the reasoning of this suggestion. Actually, we have already published a relevant study in the close hole regime of the same device in *Nano Lett.* 21, 3835–3842 (2021). In this letter, $g_{\perp} \sim 3.9$ is obtained in a transport measurement, which agrees almost perfectly with the value of $g_{\perp} = 3.95$ in this manuscript. Considering

variations due to different confinement and hole filling, to us the g-factor of $g_{\perp} = 7$ obtained for the second dot is reasonable.

Reference:

Zhang, T. et al. Anisotropic g-Factor and Spin–Orbit Field in a Germanium Hut Wire Double Quantum Dot. *Nano Lett.* **21**, 3835–3842 (2021).

About the supplementary S10

Once again, here, the only “qualitative understanding” is not sufficient for the authors to claim that the fast Rabi oscillations are only coming from an ultra-short spin-orbit length. It is known in the community that the electrically driven spin resonance of hole is a complex interplay between two mechanisms namely EDSR and g-tensor modulation (which with the g-factor reported cannot be neglected in the analysis). These two mechanism being hard to capture theoretically in the multi-hole regime with a strong LH/HH mixing, strong electrical field, possibly strain and disorder. Consequently, Figure1c cannot be used to illustrate the experimental results here as the contribution from g-tensor contribution to EDSR has not been characterized and cannot be rule-out.

Response: This is a good question. We agree that g-tensor modulation could play an important role during hole spin resonance. Therefore we have included this part on page 5 & 6 in the revised Supplementary Material, Sec. 2.

“Compared to EDSR arising from intrinsic SOC, g-tensor modulation is often also considered when studying EDSR^{S4,S5}. If we consider this effect, the effective Zeeman Hamiltonian reads

$$H_z = \frac{1}{2} \mu \sigma^T \cdot \hat{g} \cdot \mathbf{B},$$

where $\sigma = (\sigma_x, \sigma_y, \sigma_z)$ are the Pauli matrices. H_z is fully parametrized by the nine independent elements of the matrix \hat{g} . For a given $\mathbf{B} = (0, 0, B_z)$, the two hole-Hamiltonian in the five ground basis evolves into

H_{DQD}

$$= \begin{pmatrix} -\varepsilon & t/\sqrt{2} & -t/\sqrt{2} & -\Delta_{SO}^* & \Delta_{SO} \\ t/\sqrt{2} & (g_1^{33} - g_2^{33})\mu_B B/2 & 0 & -(g_1^{13} + ig_1^{23})\mu_B B/2 & -(g_2^{13} - ig_2^{23})\mu_B B/2 \\ -t/\sqrt{2} & 0 & -(g_1^{33} - g_2^{33})\mu_B B/2 & -(g_2^{13} + ig_2^{23})\mu_B B/2 & -(g_1^{13} - ig_1^{23})\mu_B B/2 \\ -\Delta_{SO} & -(g_1^{13} - ig_1^{23})\mu_B B/2 & -(g_2^{13} - ig_2^{23})\mu_B B/2 & -(g_1^{33} + g_2^{33})\mu_B B/2 & 0 \\ \Delta_{SO}^* & -(g_2^{13} + ig_2^{23})\mu_B B/2 & -(g_1^{13} + ig_1^{23})\mu_B B/2 & 0 & (g_1^{33} + g_2^{33})\mu_B B/2 \end{pmatrix}.$$

While the shear terms of the g-tensor could indeed be present in our system, they are ignored here considering the high symmetry growth direction for the hut wire^{S6}, which makes our QD in the hut wire more similar to those made from two-dimensional hole gas. We thus stick to a model of SOI driven EDSR and neglect the effects of g-factor modulation in our simulations for simplicity and clarity.”

The added References are:

- S4. Tantt, T. et al. Controlling spin-orbit interactions in silicon quantum dots using magnetic field direction. *Phys. Rev. X* **9**, 021028 (2019).
- S5. Voisin, B. et al. Electrical control of g-factor in a few-hole silicon nanowire MOSFET. *Nano Lett.* **16**, 88-92 (2016).
- S6. Zhang, T. et al. Anisotropic g-Factor and Spin–Orbit Field in a Germanium Hut Wire Double Quantum Dot. *Nano Lett.* **21**, 3835–3842 (2021).

Besides, different systems have different off-diagonal elements of g-tensor based on the nanowire growth direction and anisotropy. Indeed some systems, such as silicon nanowire (*PRL* 120,137702 (2018)), have strong effect of g-tensor modulation due to non-uniform strain while the off-diagonal elements are ignored in silicon MOS (arXiv:2012.04985 and arXiv:2003.07079). Consistent with the calculation in *PRX* 9, 021028 (2019), our result reported in *Nano Lett.* 21, 3835–3842 (2021) shows that neglecting off-diagonal terms is a reasonable approximation for our system. Last but not least, while adding a mechanism and the associated fitting parameters may give us a better fit to the experimental data, we feel that the added complexity is not warranted when considering the characteristics of the hut wire system. We hope our original figure would be helpful for readers to understand the mechanism of EDSR in a simple physical picture.

Mentioned references:

- Crippa, A. et al. Electrical spin driving by g -matrix modulation in spin-orbit qubits. *Phys. Rev. Lett.* **120**, 137702 (2018).

Liles, S. D. et al. Electrical control of the g -tensor of a single hole in a silicon MOS quantum dot. arXiv:2012.04985 (2020).

Marx, M. et al. Spin orbit field in a physically defined p type MOS silicon double quantum dot. arXiv:2003.07079 (2020).

Tanttu, T. et al. Controlling spin-orbit interactions in silicon quantum dots using magnetic field direction. *Phys. Rev. X* **9**, 021028 (2019).

Zhang, T. et al. Anisotropic g -Factor and Spin–Orbit Field in a Germanium Hut Wire Double Quantum Dot. *Nano Lett.* **21**, 3835–3842 (2021).

About figure S8

In Fig S8d, I don't understand why the simulation is not giving a gradient of field $E_y(AC)$ along the y axis. I would expect that applying the microwave signal on the right gate will make the left hole moving along the wire. Moreover it would be nice to see the $E_z(ac)$, which will participate to g -tensor modulation.

Response: We agree with the suggestion. We have revised figure S8 in Supplementary Material.

“Here, the a.c. fields along the x and y directions in Fig. S8c & d accounts for the effective driving while the z -component of the field does not contribute to EDSR (Fig. S8e). For example, at the power of $P = 0$ dBm, AC electric fields in the left (right) dot are $E_{ac}^x = 230$ V/m (715 V/m) and $E_{ac}^y = 3475$ V/m (1810 V/m) and $E_{ac} = \sqrt{(E_{ac}^x)^2 + (E_{ac}^y)^2}$ is used in the main text. Fig. S8f shows the value of a.c. field along the nanowire (i.e. y direction) and its gradient, which drives holes in both dots to move back and forth mainly along the nanowire due to the larger component along y direction. Using our dot parameters, we find a field-induced displacement of 15 pm in the right dot as shown by Fig. S8 g & h, larger than the micromagnet induced displacement of 4 pm in silicon QD^{S10,S11}.

Figure 9 and New Figure S8: **a**, Device structure from SEM image for input of size in COMSOL. **b**, Simulated z-component of static electric field E_{dc}^z after applying voltages of $V_L = 0.17$ V, $V_M = 0.085$ V, $V_R = 0.355$ V and $V_{sd} = 3$ mV. Inset: E_{dc}^z as a function of position along y direction (linecut along the dash line). A potential width of 57

nm is obtained after Gaussian fitting. **c, d & e**, Simulation results along x , y and z axes after applying an alternating electric field at $P = 0$ dBm. The white ellipse marks the position of the right dot. **f**, E_{ac}^y and dE_{ac}^y/dy as a function of position along y direction in **d, g**. The whole electric field $E^y = E_{ac}^y + E_{dc}^y$ shifts along y direction due to the microwave. We extract right-dot shift of $\Delta E^y \approx 3600$ V/m at the position underneath gate R. **h**, Using this value, we can deduce the dot displacement $\Delta y = \frac{eE_{ac}l_{dot}^2}{\hbar\omega_y}$ and the effective magnetic field $B_{ac} = 2B \cdot \frac{l_{dot}}{l_{so}} \cdot \frac{eE_{ac}l_{dot}}{\hbar\omega_y}$ due to the driving by the a.c. electric field. We find $\Delta y = 15$ pm (i.e. Δr in Figure 1c) and $B_{ac} = 3.6$ mT for the right dot spin at the driving power of $P = 0$ dBm, consistent with the 100 MHz Rabi frequency for mode B at this power with a spin-orbit length of 1.4 nm.

References:

- S10. Kawakami, E. et al. Gate fidelity and coherence of an electron spin in an Si/SiGe quantum dot with micromagnet. *PNAS* **113**, 11738-11743 (2016).
- S11. Yoneda, J. et al. A quantum-dot spin qubit with coherence limited by charge noise and fidelity higher than 99.9% *Nat. Nanotechnol.* **13**, 102-106 (2018).

“Due to the smaller a.c. field in the right dot (see Supplementary Material, Sec. 6), mode B shows a slower speed of Rabi rotation compared to mode A even though the spin orbit lengths are close”

This sentence is not complete. The left dot is also not seeing the same direction of microwave field excitation. Then the EDSR mechanism could be different.

Response: Thanks for the suggestion and we have made the revisions on page 9 in the revised main text. “Due to the smaller a.c. field in the right dot compared to the left dot (see Supplementary Material, Sec. 6), mode B shows a slower speed of Rabi rotation compared to mode A at the same microwave power even though the spin-orbit length should be close based on our calculation.”

“Moreover, a faster Rabi oscillation can be achieved by changing the inter-dot coupling or the manipulation position. Such optimization can be achieved by tuning the voltage of gate R. Our test results show that the Rabi frequency can be increased

from 63 MHz to 111 MHz by switching the manipulation position from M2 to M1 (Fig. S4 in Supplementary Material). We are thus optimistic that a faster Rabi operation is achievable after optimization.”

Between M1 and M2, the g -factor is changing, then the microwave frequency to apply to drive Rabi oscillation is different. How does the authors know that the microwave power delivered to the device is the same between the point M1 and M2. This could explain the change in Rabi frequency.

Response: In order to avoid the power variation as much as possible, we experimentally choose the regime where no step-like signal can be observed. As shown in Fig. S4b, the microwave frequency (i.e. where our qubit works) varies from 8.03 GHz to 8.05 GHz. In this regime, the zoom-in shows no step-like signal in the smooth background which means the driving microwave reaches the device with consistently similar power in the frequency range of interest to us.

Figure 10: EDSR of mode A and B ranging from 7.5 GHz to 8.5 GHz after zooming in.

To conclude, I would summarize my position as follow: Wang et al. achieve ultra-fast Rabi oscillations in a double quantum dot formed in a germanium hut wire with no clear experimental or theoretical demonstration on their origin. The resonance mechanism is only approached “qualitatively” and deep characterization of the dots

g-factor and EDSR mechanism is missing. Once again, if this manuscript would have been the first one to report spin qubits in hut wire I would support its publication (together with major corrections) but hut wire spin-qubit have already been demonstrated. Thus the lack of convincing demonstration (experimental and theoretical) on the origin of the fast Rabi oscillation reported here makes that I cannot recommend this manuscript for publication in Nature Communications.

Response: As mentioned before, compared to the work by Watzinger et al. in 2018, where a hole spin qubit in Ge is first demonstrated, our work shows much faster Rabi operation with a strong spin-orbit coupling strength arising from the different confinement and hole occupation. We report the up-to-now smallest spin-orbit length in quantum dot devices. Also, we have added a model for the complex EDSR spectrum observed, which would be useful for studying resonances in the multi-hole regime. These were not demonstrated before and should be interesting to the readers. Last but not least, ultrafast control of hole spins has also been achieved in a Ge/Si core/shell nanowire by Froning et al. (*Nature Nanotechnol.* **16**, 308-312 (2021)), though the hole configuration is very different from ours. The two systems do share the feature of a strong spin-orbit coupling, which enables fast control. We believe the dramatically faster Rabi frequency and the associated innovations do meet the requirements of novelty and importance that make the manuscript suitable for publication in Nature Communications.

Reference:

Froning, F. N. M. et al. Ultrafast hole spin qubit with gate-tunable spin-orbit switch functionality. *Nat. Nanotechnol.* **16**, 308-312 (2021).

REVIEWERS' COMMENTS

Reviewer #1 (Remarks to the Author):

The authors addressed my concerns and I am happy to recommend the manuscript for publication in its current version. I also would like to thank them for the interesting scientific discussion.

Reviewer #4 (Remarks to the Author):

I have read with great interest the manuscript entitled: "Ultrafast Coherent Control of a Hole Spin Qubit in a Germanium Quantum Dot" by Professor Guo and colleagues. I also have read carefully correspondence between the Reviewers and the Authors. The Authors study experimentally double quantum dot structure, which is electrostatically defined in gated Germanium hut nanowire hosting valence holes. The main result of the work is demonstration of fast hole spin qubit control ($f_{\text{Rabi}}=540\text{MHz}$ for 100mT) via EDSR technique with high quality factor. Such a fast Rabi oscillations indicate strong spin-orbit coupling in the system and associated short spin-orbit length. The authors also provide a theoretical model based on the effective Hamiltonian for two (heavy) holes. This allows Authors to calculate EDSR spectrum and compare the theoretical results with the experimental findings. In my opinion Authors have obtained sufficiently good matching between theoretical and experimental results.

The Reviewer #3 has several concerns about the work especially that the effective theoretical model proposed by the Authors does not describe the studied physical system and does not explain the observed EDSR spectrum, which does not match perfectly to one obtained from the effective model.

Having in mind that this is an experimental paper, I think that theoretical model proposed by the Authors provides good enough matching with the experiment. Comparing prof. Guo's manuscript with the other experimental works published in Nature group journals demonstrating hole spin qubit manipulation e.g. Refs [30, 40, 41], I can notice that none of these Refs provide a theoretical model of the studied system. Thus, I think that matching done by the Authors is an important addition to their experimental work, even though that the agreement is not perfect.

The question is; is it even possible to solve the realistic model for the studied system. First of all the Author have evidence that there is about 15 holes confined in the double quantum dot. In order to

find the eigenenergies of such a system, one would have to solve Schroedinger equation based on LK Hamiltonian for 15 interacting holes e.g. using configuration interaction (CI) method (C.-Y. Hsieh, et al, Phys. Rev. B 80, 235320 (2009); W J Pasek et al 2014 Semicond. Sci. Technol. 29 115022 (2014)). Moreover this would have to be done self consistently with Poisson equation. Due to huge computational cost, it is impossible to do it in practice. On the other hand one could solve the Poisson-Schroedinger problem just for two holes with CI method and assume that the effect of the presence of the other holes results in different g factors in the left and the right dots, which hosts different number of holes. Then, after finding the eigenstates, in order to simulate EDSR mechanism, one would have to solve time dependent Schroedinger equation, which takes into account the effect of "shaking" of one of the holes, which is induced by oscillating voltage applied to one of the gates. Such an approach could give more realistic and accurate results than the one obtained within the effective model. However, in my opinion, this is far beyond the expertise of most of the experimentalists and this is a subject for a separate theoretical work. I think that in the case of experimental paper the effective model can give more qualitative understanding of the physics of the system than complicated numerical calculations.

Below I refer to some other concerns raised by the Referee #3:

(i)

A: "* with a strong spin-orbit coupling strength arising from the very different confinement thanks to the additional middle gate induced tunable inter-dot coupling.*"

R#3:"* There is no experimental data indicating that the confinement is largely changed compared to Watzinger et al. thanks to the middle gate. *"

In my opinion the double quantum dot set-up fabricated by the Authors is greatly improved in terms of size and functionality comparing to the one from Watzinger et al work. There is no doubt that such configuration of gates and applied voltages gives better control over the system in terms of confinement potential. The middle gate allows for tuning the inter-dot coupling strength (exchange interaction) and allows applying positive voltages to the gates L and R. From my expertise for such narrow gates and high voltages, I would expect to that the confinement potential is stronger and the dots size are smaller comparing to the Watzinger et al. work. However, solving Poisson equation for both the systems, would give the definitive answer. In fact, for completeness the Authors could add the plot of confinement in the y direction coming from the electrostatic potential distribution along y-axis obtained by Comsol calculations in section 6 of SM.

Looking at the Fig 1. B we can see that voltages applied to the gates V1 and V2 are different by about 100meV which in principle can produce electric field gradient which can affect and improve the SO strength.

(ii)

A: "*Moreover, a faster Rabi oscillation can be achieved by changing the inter-dot coupling*"

R#3: "*Could the authors argue about this point? Because the Rabi frequency presented here are measured deep in Coulomb Blockade for which in principle single spin transition are considered (as claimed by the authors themselves), then the interdot coupling does not play any role here. The statement is to me wrong.*"

Authors have corrected an erroneous statement and this should be sufficient to accept the response.

(iii)

A: "*The ultrafast spin control we have demonstrated suggests a strong potential of hut wire Ge hole spin qubits for scalable high-fidelity qubit control.*"

R#3: "*The verb "suggest" indicates effectively that nothing in this manuscript is demonstrating high fidelity...*"

I think that precise fabrication of very small gates deposited on 2nm "high" hut nanowire and demonstration of the EDSR driven fast hole qubit manipulation together with quality factor is a very promising step towards realization of scalable electrically controlled basic elements of quantum computing architecture.

(iv)

R#3: "*The authors present a simulated EDSR spectrum in order to explain their data. For me there is two points I would like to raise here:*

1. In the figure below (bottom left) I superimposed colored lines on the experimental data and then did a mirror along the vertical axis in order to compare the experimental EDSR lines I see and the calculated spectrum. I can agree with the authors that the spectrum looks close but still there is some discrepancy. 2 EDSR lines totally not captured by the simulation (black dashed lines) and anticrossing not captured also (between red and yellow lines close to zero field and around 6 GHz between the 3 EDSR lines visible in the experiment, only two in the experiment)."

The authors added two excited states into their effective Hamiltonian, which in my opinion gives better-fit than original ground state model to the experimental data i.e. appearance of two missing EDSR lines. I think that as for an experimental study this is sufficient theoretical explanation of the observed results.

(v)

R#3: "From all the above remarks, I do not understand how the authors can claim (even qualitatively) that they are driving an HH spin transition. From the complexity of the EDSR spectrum, it appears clear that the level splitting in one of the QD (at least) is small. From the small g-factor, it is also clear that the states involved in the EDSR resonances have mixed heavy and light hole character. The Rabi frequency observed is then certainly coming from a complex interplay between spin-orbit interaction and multi-level structure with heavy and light admixture of the QD under consideration. Consequently, I cannot agree with the authors on the description of the EDSR mechanism as just the effect of a Rashba-type SOI acting on spin 3/2 heavy hole."

From my (theoretical) expertise for such high anisotropy between confinement in the z direction which is much smaller than the one in (x,y) plane the holes are mainly in the HH state. However some other effects mentioned by the Reviewer#3 can induce HH-LH mixing. In my opinion the indirect proof of dealing with the HHs is the relatively long coherence time. On the other hand, even if there is a significant amount of mixing between the HH and the LH states, the Authors managed to show fast and coherent transitions between two quantum states, which is needed for a good candidate for a qubit and in my opinion this is very promising experimental achievement. Maybe some optical experiments for such a system could tell more about the nature of the states in the dots and HH-LH mixing.

After reading the manuscript I have some additional questions and remarks.

1) In the supplementary material in the Section 9 the Authors refer to inset of Fig S9b, which I couldn't find. Is the inset missing or did Authors wanted to refer to inset of an other Fig ?

2) The Authors showed theoretically weak HH-LH mixing based on diagonalization of Hamiltonian from Section 9 with hard wall boundary conditions. I believe that for taken $L_x=20\text{nm}$, $L_y=57\text{nm}$ and $L_z=2\text{nm}$ the ground states is mainly composed of HH (98%) as shown by the Authors. However if we add a harmonic like confinement potential in (x,y) plane as in Section 10 with $l_{\text{dot}}=5\text{nm}$ (line 242

and 244 from the main text) I can expect that this can give a stronger in plane confinement (and weaker anisotropy between the inplane and the z direction confinement) than the hard wall boundary conditions and consequently contribution of LH might be bigger than 2%. The best way to answer that question would be to take the electrostatic potential distribution in y direction obtained from Comsol calculations (the confinement in the x can be hard wall) and put it into Hamiltonian and then diagonalize the Hamiltonian matrix.

3) I think it would be valuable for theorists who would like to study such a system numerically to add a cross-section of the device (in the x-z plane) including nanowire hut as in Fig S1a, but with an approximate size of each layer and electrodes and the distance between the gates given in nanometers.

To conclude, in my opinion the Authors have adequately addressed the issues raised by the Referees. The paper is self-contained, provides in depth analysis and discussion of obtained results and comparison with other similar works. Having in mind that the hole spin qubits are very promising platform for scalable quantum computation, and that the Authors have managed to demonstrate fast hole spin qubit manipulation together with relatively high quality factor, I believe that obtained results, even though that not perfect, will stimulate further progress in the field of hole spin qubits in Germanium hut nanowires with application to scalable quantum computation. I think that the results are interesting, important and provide sufficient novelty to meet the Nature Communications standards.

Reviewer #3 (Remarks to the Author):

I appreciate the efforts that Wang et al. did to improve their manuscript. However, I still maintain my position about this work.

Here I summary the issues justifying my position:

- The experimental QD are far from what the author claim with their HH modeling with the g-factor reported. The authors argue that taking into account all the elements which can change the hole nature is difficult and beyond the scope of this work and I agree with them. However, unfortunately all the physics is there, and claiming that the 500MHz of Rabi frequency achieved here is understood with a simple model is not right.
- The authors have now to invoke an excited orbital state at 12microeV from the ground state to explain their EDSR spectrum. This was exactly my concern in the three first paragraphs of my first review. This small level splitting is then certainly one of the major origin of the fast Rabi oscillations reported and not the exceptionally small spin-orbit length reported by the authors.
- The spin-orbit length analysis has to be changed to take into account this small level splitting.
- Even if I understand that the characterization of the g-tensor modulation is beyond the scope of the manuscript. I disagree when the authors argue that their QDs are similar to those made from 2D hole gas which justify to stick to a model of SOI driven EDSR. Once again, I would agree if the reported g-factors would go in this direction. Unfortunately, this is not the case here suggesting that the g-tensor modulation cannot be neglected in the analysis of the origin of the fast Rabi oscillation.

To conclude, I would summarize my position as follow: Wang et al. achieve ultra-fast Rabi oscillations in a double quantum dot formed in a germanium hut wire with no clear experimental or theoretical demonstration on their origin.

Once again, if this manuscript would have been the first one to report spin qubits in hut wire I would support its publication but hut wire spin-qubit have already been demonstrated. Thus the lake of convincing demonstration (experimental and theoretical) on the origin of the fast Rabi oscillation reported here makes that I cannot recommend this manuscript for publication in Nature Communications.

Response to Reviewers' Comments

We thank the Reviewers for their time and patience with our manuscript. Below we give a point by point answer to the questions raised by the Reviewers. The comments from the Reviewers are reproduced in *purple color and Calibri font* along with our responses in black text while modified contents are indicated by *blue color and Times New Roman font*.

Response to Reviewer 1

Comments: The authors addressed my concerns and I am happy to recommend the manuscript for publication in its current version. I also would like to thank them for the interesting scientific discussion.

Response: We would like to thank the reviewer for his/her continued efforts, and we are grateful for this positive recommendation.

Response to Reviewer 3

*Comments: I appreciate the efforts that Wang et al. did to improve their manuscript. However, I still maintain my position about this work. Here I summary the issues justifying my position:
- The experimental QD are far from what the author claim with their HH modeling with the g-factor reported. The authors argue that taking into account all the elements which can change the hole nature is difficult and beyond the scope of this work and I agree with them. However, unfortunately all the physics is there, and claiming that the 500MHz of Rabi frequency achieved here is understood with a simple model is not right.*

Response: We thank the Reviewer for appreciating our effort in improve the manuscript. However, we respectfully disagree with the referee's assertion that our model does not catch the essence of the physics in this experiment. We argue that our model, while relatively simple and is probably not the full picture, does contain the essential ingredients of the experimental results. First, our model mapped the resonance pattern with quite a

good match. Second, we conclude that the very small spin-orbit length in our system hints at a strong spin-orbit coupling, which facilitates the ultrafast operations we observed, and we believe this is a reasonable assertion when compared with Froning et al.'s recent work (*Nature Nanotechnol.* **16**, 308-312 (2021)). We do acknowledge that spin-orbit length is a subtle concept in a quantum dot where there is no transport and hole momentum averages to zero. Nevertheless, we include it in our discussion because it is commonly used in other experimental studies and is thus a useful parameter when comparing different devices. As agreed by the reviewer, a thorough and realistic theoretical study of the hole spectrum and states is a difficult task and is beyond the scope of this work. Thus, we believe our model provides a concise and qualitatively correct physical picture for our system, and is overall a positive addition in the analysis of our experimental finds, even if the matching is not 100% perfect.

- The authors have now to invoke an excited orbital state at 12microeV from the ground state to explain their EDSR spectrum. This was exactly my concern in the three first paragraphs of my first review. This small level splitting is then certainly one of the major origin of the fast Rabi oscillations reported and not the exceptionally small spin-orbit length reported by the authors.

Response: We understand the reviewer's argument and agree that excited states with small energy splitting may indeed induce faster EDSR. Please note that spin-orbit coupling $\alpha_2 \approx \frac{1}{6} \alpha_{eff} \cdot a_x^2$ with $l_{so} \approx \frac{\hbar^2}{m\alpha_{eff}}$ (as mentioned in Supplementary Note 10), while $f_{Rabi} \approx \frac{g\mu_B B}{h} \cdot \frac{6\alpha_2 \cdot m \cdot eE_{ac}}{\hbar^3 \omega_y}$ according to Eq. 2 in our main text. This means that a small confinement length along x , together with a strong spin-orbit coupling α_2 , would lead to a small spin-orbit length, while strong spin-orbit coupling, together with Zeeman splitting and nearby intermediate state(s) (the Rabi formula above is derived for a single hole, for which $\hbar\omega_y$ is the excitation energy), are main factors for faster control. While spin-orbit coupling strength α_2 underpins both quantities, additional factors affect both as well. As such it is not accurate to assert that small spin-orbit length is the reason for the fast Rabi oscillation. We have thus modified the text and the Supplementary Notes to emphasize the importance of spin-orbit coupling strength (instead of spin-orbit length) for the fast Rabi oscillations we observed.

An important point raised by the referee here is whether the nearby excited state we include in our model is the determining factor in the ultrafast Rabi oscillation we have observed. We believe that while this excited state may indeed have contributed to the faster of the Rabi oscillations we have seen (mode A), it is still the strong spin-orbit coupling that is the determining factor. The robustness of the spin blockade in our system clearly indicates that the excited state is localized in one of the dots (more specifically the empty one in the (02) charge configuration). Otherwise the spin blockade would be easily lifted by this low-energy excited state. As such only mode A EDSR signal is enhanced by this intermediate state, while mode B is not be affected. Therefore, the fact that our mode B Rabi frequency can also reach a very fast 290 MHz is a clear indication that the underlying spin-orbit coupling has to be very strong, while the even faster Rabi oscillation of mode A is probably enhanced by the low-energy excited state.

Again, we thank the referee for raising this really pertinent question and pushing us to be more precise in our statements, and we have made additions and modifications in the main text and the Supplementary Notes 2 and 10 to clarify this issue, and to make more physically sound attributions about the origins of our observed fast Rabi oscillations.

As such, we added the sentence on page 7 in the main text:

“We thus attribute the large Rabi frequencies of both mode A and B to a large value of α_2 . The particularly large Rabi frequency in mode A may have been enhanced by the low-energy excited state included in our model (Supplementary Note 2 and 10), a fact that has previously been exploited to enhance spin-electric-field coupling^{50,51}.”

And added the sentences in the discussion on page 8:

“We report a small spin-orbit length in a smaller Ge double quantum dot compared to existing work on GHW in the literature⁴⁰ with narrower electrodes. We attribute the ultrafast control of a hole spin qubit that we have observed to an overall strong spin-orbit coupling, possibly assisted by a nearby excited state, even though the relative smaller longitude dot size (along y) may have reduced the Rabi frequency.”

50. Hu, X., Liu, Y. X. & Nori, F. Strong coupling of a spin qubit to a superconducting stripline cavity. *Phys. Rev. B* **86**, 035314 (2012).

51. Croot, X. et al. Flopping-mode electric dipole spin resonance. *Phys. Rev. Research* **2**, 012006 (2020).

For clarity, we also added the discussion in Supplementary Note 2 and Supplementary Note 10:

“While in this model we do not assume any particular characteristics for the involved orbital states, the robust spin blockade we have observed is a strong indication that the excited orbital involved in the (1,1) excited state is localized in the “empty dot” side of the (0,2) configuration. If it had been localized in the blocked dot (the “2” side of the (0,2) configuration), it would provide a low-energy triplet that lifts the spin blockade.”

“Note that fast control of hole spin arising from strong spin-orbit coupling which is not only related to the small spin-orbit length but also determined by other effects, such as transverse size or excited states with small energy splitting. However, as we discussed in Supplementary Note 2, we believe the low-energy excited states that are present in our double dot is localized in one of the two dots, such that it would only enhance the EDSR Rabi frequency in that dot. The fact that Rabi frequencies in both mode A and mode B are high (540 MHz and 290 MHz at 9 dB driving power) is thus a likely consequence that our system has strong spin-orbit coupling strength (thus short l_{so}).”

- The spin-orbit length analysis has to be changed to take into account this small level splitting.

Response: We obtain the spin-orbit length according to Eq. 2 in main text for a single hole. As shown in that equation, the spin-orbit length also depends on both the spin-orbit coupling strength and the transverse size of the quantum dot. In other words, short spin-orbit length and fast Rabi oscillation are not equivalent as each is affected by additional factors beyond spin-orbit coupling, although a strong spin-orbit coupling could lead to both. In other words, short spin-orbit length and fast Rabi oscillation do not have a causal relationship. Instead they could both be consequences of a strong spin-orbit coupling.

Underlying relationship notwithstanding, we believe that the spin-orbit length of 1.5 nm obtained from our calculation is reasonable in this type of quantum dot because it is consistent with published works, such as the ultrafast control in core/shell Ge nanowire

even though their device is quite from ours (*Nature Nanotechnol.* **16**, 308-312 (2021)) where a spin-orbit length of 3 nm is obtained in the multi-hole regime as well.

Although spin-orbit length and EDSR Rabi frequency do not have a simple one-to-one relationship, we would like to keep our spin-orbit length analysis in the manuscript, mostly because it gives a parameter that is commonly used to represent the strength of spin-orbit coupling in a sample, therefore allowing comparisons of different devices and materials.

We have added clarifications on page 6 in the main text and removed any misleading statement.

“Notice that while spin-orbit length is a concept more appropriate for free carriers, it is nonetheless useful for comparing different systems. For example, for strongly spin-orbit coupled conduction electrons confined in InAs quantum dots of similar size to our dots, typical spin-orbit length ranges from 100 to 200 nm⁴⁹. In comparison, our hole system has, inherently, a much stronger spin-orbit coupling (thus a much smaller l_{so}), which is the determining factor for the ultrafast operation of our qubit.”

- Even if I understand that the characterization of the g-tensor modulation is beyond the scope of the manuscript. I disagree when the authors argue that their QDs are similar to those made from 2D hole gas which justify to stick to a model of SOI driven EDSR. Once again, I would agree if the reported g-factors would go in this direction. Unfortunately, this is not the case here suggesting that the g- tensor modulation cannot be neglected in the analysis of the origin of the fast Rabi oscillation.

Response: As mentioned before, our result reported in *Nano Lett.* 21, 3835–3842 (2021) shows that neglecting off-diagonal terms of g-tensor is a reasonable approximation for our system. The hut wire has a trapezoidal cross-section that is 1.5 nm in height, 5 nm wide on the top and 20 nm wide at the bottom, and has high degree of left-right symmetry. With our gate geometry, our quantum dots should be located near the center of the nanowire, giving them a highly symmetric environment in terms of strain/lattice distortion, making it much more similar to 2D quantum dot than, for example, a corner dot in a Si nanowire as reported in *PRL*. 120,137702 (2018). In other words, the off-diagonal terms for the g-tensor should be much smaller than the diagonal ones, which means the g-tensor modulation, i.e. the variation of the off-diagonal terms, should be small.

While the g-factor for our two quantum dots are quite different (interdot distance in the order of 50 nm,), the difference there does not have a direct influence on the EDSR as the signals we observed are mostly single-dot effects, when the dot movement is in the order of 15 pm driven by the largest electric field we applied. Thus for g-factor modulation induced EDSR we need to examine g-factor variation over a distance in the order of tens of pm.

While g-factor modulation can certainly be included in our theoretical description, it is difficult to estimate quantitatively without full knowledge of the interface roughness profile, which is critical factor (in addition to the spin-orbit coupling strength) to EDSR by g-factor modulation. Considering the highly symmetric and relatively uniform strain profile our hut wire should have along the y-direction, we do not believe strain would play a large role in g-factor variations when the AC electric field is applied. The length scale characterising interface roughness is a few Angstroms [see Phys. Rev. B 82, 205315 (2010) and references therein]. Roughness can be included by redefining the location of the interface from $z = 0$ to $z = \zeta(x,y)$, where $\zeta(x,y)$ is a random function of x and y . Working as we do here in terms of the basis functions for $\zeta = 0$, one can consider $V(z - \zeta) - V(z)$ as an effective potential characterising roughness [$V(z)$ being the confinement potential in the z -direction]. In the case of EDSR the term $V(z - \zeta) - V(z)$ will depend on the ac field E (which determines the range of dot displacement), and will play the same role that eEx currently plays in our model. From the estimated dot movement of 15 pm under the action of the electric field, it is possible that roughness makes a small contribution to the EDSR Rabi frequency, though without full knowledge of the interface and barrier it is difficult to make a meaningful estimate.

Based on the above discussion, we conclude that the most likely source of EDSR in our setup is the strong spin-orbit interaction, either acting directly on the electron, or in some small part via g-factor modulation. We wish to stress, however, that g-factor modulation does NOT occur in the absence of the spin-orbit interaction, therefore, although the final numbers may differ slightly when g-factor modulation is incorporated, our measurements provide convincing evidence of strong spin-orbit coupling in our samples.

To conclude, I would summarize my position as follow: Wang et al. achieve ultra-fast Rabi oscillations in a double quantum dot formed in a germanium hut wire with no clear experimental or theoretical demonstration on their origin.

Once again, if this manuscript would have been the first one to report spin qubits in hut wire I would support its publication but hut wire spin-qubit have already been demonstrated. Thus the lake of convincing demonstration (experimental and theoretical) on the origin of the fast Rabi oscillation reported here makes that I cannot recommend this manuscript for publication in Nature Communications.

Response: As mentioned before, compared to the work by Watzinger et al. in 2018, where a hole spin qubit in Ge is first demonstrated, our work shows much faster Rabi operation with a strong spin-orbit coupling strength arising from the different confinement and hole occupation. To answer the question of the origin of fast control, we thus added the sentences in the discussion on page 8:

“We report a small spin-orbit length in a smaller Ge double quantum dot compared to existing work on GHW in the literature⁴⁰ with narrower electrodes. We attribute the ultrafast control of a hole spin qubit that we have observed to an overall strong spin-orbit coupling, possibly assisted by a nearby excited state, even though the relative smaller longitude dot size (along y) may have reduced the Rabi frequency.”

Response to Reviewer 4

Comments: I have read with great interest the manuscript entitled: "Ultrafast Coherent Control of a Hole Spin Qubit in a Germanium Quantum Dot" by Professor Guo and colleagues. I also have read carefully correspondence between the Reviewers and the Authors. The Authors study experimentally double quantum dot structure, which is electrostatically defined in gated Germanium hut nanowire hosting valence holes. The main result of the work is demonstration of fast hole spin qubit control ($f_{\text{Rabi}}=540\text{MHz}$ for 100mT) via EDSR technique with high quality factor. Such a fast Rabi oscillations indicate strong spin-orbit coupling in the system and associated short spin-orbit length. The authors also provide a theoretical model based on the effective Hamiltonian for two

(heavy) holes. This allows Authors to calculate EDSR spectrum and compare the theoretical results with the experimental findings. In my opinion Authors have obtained sufficiently good matching between theoretical and experimental results.

Response: We are very grateful to referee's efforts and strong supportive statements.

The Reviewer #3 has several concerns about the work especially that the effective theoretical model proposed by the Authors does not describe the studied physical system and does not explain the observed EDSR spectrum, which does not match perfectly to one obtained from the effective model.

Having in mind that this is an experimental paper, I think that theoretical model proposed by the Authors provides good enough matching with the experiment. Comparing prof. Guo's manuscript with the other experimental works published in Nature group journals demonstrating hole spin qubit manipulation e.g. Refs [30, 40, 41], I can notice that none of these Refs provide a theoretical model of the studied system. Thus, I think that matching done by the Authors is an important addition to the their experimental work, even though that the agreement is not perfect.

Response: We agree our model may be not perfect but believe that it can be used to explain our experimental observation. We appreciate the referee for pointing out the originality of our theoretical model.

The question is; is it even possible to solve the realistic model for the studied system. First of all the Author have evidence that there is about 15 holes confined in the double quantum dot. In order to find the eigenenergies of such a system, one would have to solve Schroedinger equation based on LK Hamiltonian for 15 interacting holes e.g. using configuration interaction (CI) method (C.-Y. Hsieh, et al, Phys. Rev. B 80, 235320 (2009); W J Pasek et al 2014 Semicond. Sci. Technol. 29 115022 (2014)). Moreover this would have to be done self consistently with Poisson equation. Due to huge computational cost, it is impossible to do it in practice. On the other hand one could solve the Poisson-Schroedinger problem just for two holes with CI method and assume that the effect of the presence of the other holes results in different g factors in the left and the right dots, which hosts different number of holes. Then, after finding the eigenstates, in order to simulate EDSR mechanism, one would have to solve time dependent Schroedinger equation, which takes into account the effect of "shaking" of one of the holes, which is induced by oscillating voltage applied to one of the gates. Such an approach could give more realistic and

accurate results than the one obtained within the effective model. However, in my opinion, this is far beyond the expertise of most of the experimentalists and this is a subject for a separate theoretical work. I think that in the case of experimental paper the effective model can give more qualitative understanding of the physics of the system than complicated numerical calculations.

Response: We agree that the specific study of the theory is very difficult especially with so many elements that may have effects on the EDSR spectrum. We thank the reviewer to point out that our model is effective enough to support the understanding of the system.

Below I refer to some other concerns raised by the Referee #3:

(i)

A: " with a strong spin-orbit coupling strength arising from the very different confinement thanks to the additional middle gate induced tunable inter-dot coupling."

R#3:" There is no experimental data indicating that the confinement is largely changed compared to Watzinger et al. thanks to the middle gate. "

In my opinion the double quantum dot set-up fabricated by the Authors is greatly improved in terms of size and functionality comparing to the one from Watzinger et al work. There is no doubt that such configuration of gates and applied voltages gives better control over the system in terms of confinement potential. The middle gate allows for tuning the inter-dot coupling strength (exchange interaction) and allows applying positive voltages to the gates L and R. From my expertise for such narrow gates and high voltages, I would expect to that the confinement potential is stronger and the dots size are smaller comparing to the Watzinger et al. work. However, solving Poisson equation for both the systems, would give the definitive answer. In fact, for completeness the Authors could add the plot of confinement in the y direction coming from the electrostatic potential distribution along y-axis obtained by Comsol calculations in section 6 of SM.

Looking at the Fig 1. B we can see that voltages applied to the gates V1 and V2 are different by about 100meV which in principle can produce electric field gradient which can affect and improve the SO strength.

Response: We want to thank the reviewer for pointing out the difference between our work and the one by Watzinger et al. These narrower electrodes of our device are indeed smaller but we obtained the result of fast control which should thank to the strong spin-orbit coupling strength α_2 as mentioned above. To make the story complete, we have followed the suggestion of adding the electrostatic potential distribution along y-axis and we have thus inserted this plot in Supplementary Note 6.

Revised Supplementary Figure 8b: **b**, Simulated y -component of static electric field E_{dc}^y after applying voltages of $V_L = 0.17$ V, $V_M = 0.085$ V, $V_R = 0.355$ V and $V_{sd} = 3$ mV. Inset: E_{dc}^y and obtained electrostatic potential distribution $V(y)$ as a function of position along y direction (linecut along the dash line).

(ii)

A: "Moreover, a faster Rabi oscillation can be achieved by changing the inter-dot coupling"

R#3: " Could the authors argue about this point? Because the Rabi frequency presented here are measured deep in Coulomb Blockade for which in principle single spin transition are considered (as claimed by the authors themselves), then the interdot coupling does not play any role here. The statement is to me wrong."

Authors have corrected an erroneous statement and this should be sufficient to accept the response.

Response: We are sorry to make this error even though we have corrected it. We want to express our thanks for reviewer's support.

(iii)

A: " The ultrafast spin control we have demonstrated suggests a strong potential of hut wire Ge hole spin qubits for scalable high-fidelity qubit control."

R#3: " The verb "suggest" indicates effectively that nothing in this manuscript is demonstrating high fidelity..."

I think that precise fabrication of very small gates deposited on 2nm "high" hut nanowire and demonstration of the EDSR driven fast hole qubit manipulation together with quality factor is a very promising step towards realization of scalable electrically controlled basic elements of quantum computing architecture.

Response: We are grateful for reviewer's positive statement.

(iv)

R#3: "The authors present a simulated EDSR spectrum in order to explain their data. For me there is two points I would like to raise here:

1. In the figure below (bottom left) I superimposed colored lines on the experimental data and then did a mirror along the vertical axis in order to compare the experimental EDSR lines I see and the calculated spectrum. I can agree with the authors that the spectrum looks close but still there is some discrepancy. 2 EDSR lines totally not captured by the simulation (black dashed lines) and anticrossing not captured also (between red and yellow lines close to zero field and around 6 GHz between the 3 EDSR lines visible in the experiment, only two in the experiment)."

The authors added two excited states into their effective Hamiltonian, which in my opinion gives better-fit than original ground state model to the experimental data i.e. appearance of two missing EDSR lines. I think that as for an experimental study this is sufficient theoretical explanation of the observed results.

Response: We are delighted that the reviewer thinks our theoretical explanation is sufficient to support our experiment.

(v)

R#3: "From all the above remarks, I do not understand how the authors can claim (even qualitatively) that they are driving an HH spin transition. From the complexity of the EDSR spectrum, it appears clear that the level splitting in one of the QD (at least) is small. From the small g-factor, it is also clear that the states involved in the EDSR resonances have mixed heavy and light hole character. The Rabi frequency observed is then certainly coming from a complex interplay between spin-orbit interaction and multi-level structure with heavy and light admixture of the QD under consideration. Consequently, I cannot agree with the authors on the description of the EDSR mechanism has just the effect of a Rashba-type SOI acting on spin 3/2 heavy hole."

From my (theoretical) expertise for such high anisotropy between confinement in the z direction which is much smaller than the one in (x,y) plane the holes are mainly in the HH state. However some other effects mentioned by the Reviewer#3 can induce HH-LH mixing. In my opinion the indirect proof of dealing with the HHs is the relatively long coherence time. On the other hand, even if there is a significant amount of mixing between the HH and the LH states, the Authors managed to show fast and coherent transitions between two quantum states, which is needed for a good candidate for a qubit and in my opinion this is very promising experimental achievement. Maybe some optical experiments for such a system could tell more about the nature of the states in the dots and HH-LH mixing.

Response: We agree with the reviewer and believe that the holes are mainly in the HH state which is supported by our calculation in Supplementary Note 9.

After reading the manuscript I have some additional questions and remarks.

1) In the supplementary material in the Section 9 the Authors refer to inset of Fig S9b, which I couldn't find. Is the inset missing or did Authors wanted to refer to inset of another Fig ?

Response: Sorry for the typo and it should be the Fig S8b. We have thus corrected it in Supplementary Note 9 and replaced the trivial E_{dc}^z with E_{dc}^y following the reviewer's suggestion as mentioned above.

“Using a magnetic field of 100 mT (close to mode A and B) along the z direction as in the experiment, $L_x = 5$ nm, $L_y = 40$ nm from Comsol simulation (inset of Supplementary Fig. 8b), $L_z \leq 2$ nm and other parameters,”

Revised Supplementary Figure 8b: **b**, Simulated y -component of static electric field E_{dc}^y after applying voltages of $V_L = 0.17$ V, $V_M = 0.085$ V, $V_R = 0.355$ V and $V_{sd} = 3$ mV. Inset: E_{dc}^y and obtained electrostatic potential distribution $V(y)$ as a function of position along y direction (linecut along the dash line).

2) The Authors showed theoretically weak HH-LH mixing based on diagonalization of Hamiltonian from Section 9 with hard wall boundary conditions. I believe that for taken $L_x=20$ nm, $L_y=57$ nm and $L_z=2$ nm the ground states is mainly composed of HH (98%) as shown by the Authors. However if we add a harmonic like confinement potential in (x,y) plane as in Section 10 with $l_{dot} = 5$ nm (line 242 and 244 from the main text) I can expect that this can give a stronger in plane confinement (and weaker anisotropy between the inplane and the z direction confinement) than the hard wall boundary conditions and consequently contribution of LH might be bigger than 2%. The best way to answer that question would be to take the electrostatic potential distribution in y direction obtained from Comsol calculations (the confinement in the x can be hard wall) and put it into Hamiltonian and then diagonalize the Hamiltonian matrix.

Response: This a very good point. We have thus put the electrostatic potential distribution, as displayed in the inset of Figure S8b, into Hamiltonian and obtained a 95% HH ground state which indeed agree with the reviewer’s prediction.

“Using a magnetic field of 100 mT (close to mode A and B) along the z direction as in the experiment, $L_x = 5$ nm, $L_y = 40$ nm from Comsol simulation (inset of Supplementary Fig. 8b), $L_z \leq 2$ nm and other parameters, we diagonalize the (108×108) matrix of the Hamiltonian, and obtain a spin expectation value of $\langle J_z \rangle \approx 1.45$, from which we find the probability of HH to be $p_{HH} \approx 95\%$ from $\frac{3}{2} p_{HH} + \frac{1}{2} (1 - p_{HH}) \approx 1.45$.”

In addition, we think we may have confused the reviewer by noting the transverse dot size as l_{dot} . To improve the readability of our manuscript, we have revised this notation in the main text and Supplementary Information:

$$hf_{\text{Rabi}} = g\mu_B B \cdot \frac{a_x}{l_{\text{so}}} \cdot \frac{eE_{ac}a_x}{\hbar\omega_y}, \quad (2)$$

where $a_x = \sqrt{\hbar/(m^*\omega_x)}$ is the transverse QD size,”

“To this end, considering the wire transport direction parallel to y axis, we may replace k_x^2 in H_{so} by its average $\langle k_x^2 \rangle \propto 1/(a_x^2)$. Next, we retain only the leading order term in k_y , which we can write as $H_{so}^{1D} \approx \alpha_{eff}\sigma_x k_y$, with $\alpha_{eff} \approx \frac{6\alpha_2}{a_x^2}$.”

3) I think it would be valuable for theorists who would like to study such a system numerically to add a cross-section of the device (in the x-z plane) including nanowire hut as in Fig S1a, but with an approximate size of each layer and electrodes and the distance between the gates given in nanometers.

Response: Following reviewer’s suggestion, we have added the cross-section in Figure S1a and the sizes in the caption.

Revised Supplementary Figure 1a: **a**, Schematic representation of the three-gate device and the cross-section along the dash line. The GHW, which consists of a Si cap and a Ge layer, is grown on Si substrate and covered by a layer of aluminum oxide. Three 35 nm wide gates, spaced at 30 nm, are deposited on top of this insulating layer and HW.

To conclude, in my opinion the Authors have adequately addressed the issues raised by the Referees. The paper is self-contained, provides in depth analysis and discussion of obtained results and comparison with other similar works. Having in mind that the hole spin qubits are very promising platform for scalable quantum computation, and that the Authors have managed to demonstrate fast hole spin qubit manipulation together with relatively high quality factor, I believe that obtained results, even though that not perfect, will stimulate further progress in the field of hole spin qubits in Germanium hut nanowires with application to scalable quantum computation. I think that the results are interesting, important and provide sufficient novelty to meet the Nature Communications standards.

Response: In conclusion, we are very grateful for the strong support the reviewer has given us to publish the paper. The recommendations indeed help us to enhance the theoretical part and improve the quality of our work.